# Essential Material Knowledge and Recent Model Developments for REBCO-Coated Conductors in Electric Power Systems

**DOI:** 10.3390/ma14081892

**Published:** 2021-04-10

**Authors:** Frederic Trillaud, Gabriel dos Santos, Guilherme Gonçalves Sotelo

**Affiliations:** 1Instituto de Ingeniería, Universidad Nacional Autónoma de México, Ciudad de Mexico 04350, Mexico; 2Departamento de Engenharia Elétrica (TEE), Universidade Federal Fluminense (UFF), Niterói 21240-210, Brazil; gdsantos@id.uff.br (G.d.S.); gsotelo@id.uff.br (G.G.S.)

**Keywords:** 2G HTS wire, Finite Element Method, lumped-parameters models, power systems engineering

## Abstract

The manufacturing of commercial REBCO tapes, REBCO referring to Rare-earth barium copper oxide, has matured enough to lead to a variety of applications ranging from scientific instruments to electric power systems. In particular, its large current density with a high *n* index and low hysteresis losses make it a strong candidate for specific applications relying on the dependence of its resistance on current. Despite its advantages, there are still issues that remain to be addressed, such as the scarcity of experimental data for the basic characteristics of the superconductor over a wide range of temperature and applied magnetic field, the inhomogeneity of these characteristics along the conductor length, as well as the anisotropy of the critical current and *n* index with respect to the direction of the applied magnetic field. To better utilize the technology, it is therefore sensible to understand the relevancy of these issues so that one could simulate as accurately as possible the physics of the superconductor, at least the dynamics that may impact the correct operation of the superconducting device. There are different levels of modelling to achieve such a goal that can either focus on the performance of the superconductor itself, or on the whole device. The present work addresses some of the latest developments in the modelling of commercial REBCO tapes in power systems with a particular focus on the thermoelectric behavior of superconducting devices connected to external circuits. Two very different approaches corresponding to two different scales in the modelling of superconducting devices are presented: (1) analysis using equivalent models and lumped parameters to study the thermoelectric response of superconducting devices as a whole, (2) Finite Element Analysis (FEA) to compute distributed fields such as current density, magnetic flux density and local losses in tapes. In this context, this paper reviews both approaches and gives a broad variety of examples to show their practical applications in electric power systems. Firstly, they show the relevance of the technology in power systems engineering. Secondly, they allow inferring the necessary level of model details to optimize the operation of superconducting power devices in power grids. This level of details relies completely on the knowledge of some basic measurable properties of superconducting tapes (critical current and *n* index) and their cooling conditions.

## 1. Introduction

Electrical equipment using High-Temperature Superconducting (HTS) materials are reaching maturity after more than two decades of research and development. Nowadays, devices like Fault-Current Limiters (FCL) [1,2,3,4,5,6], power cables [7,8], and electric machines [9,10] have been built, and in some cases, installed in electric networks. The HTS materials with the most potential to address simultaneously cost and technical requirements are the second generation (2G) wire (or tape). The 2G wires are also known as REBCO tapes or coated conductors. Several manufacturers are producing high-quality 2G wires of hundreds of meters long without electrical splices [11]. They are made by depositing a thin superconducting ceramic film (from 1 to 2 μm thick) on a structured substrate. As HTS materials present a high non-linear behavior with respect to current, temperature and magnetic field, their modelling has always been challenging. In this context, this work presents some of those challenges and introduces the most recent developments in the modelling of commercial REBCO tapes for applications in power systems. In particular, it deals with systems for which an inductive or non-inductive coil has to be modelled as part of the superconducting power device [12,13]. To this date, different ways of modelling superconducting components have been devised to unveil their thermal and electrical behavior during transient operations and to look at their impact on power systems. Purely phenomenological models assuming time-dependent resistance switching from zero to a saturated value have been largely employed in power systems engineering studies [14,15]. However, these models do not include the actual physics of the superconductors. One may have to rely on experimental data to provide an accurate account of the behavior of superconducting devices. Over the past ten years, new models have been developed to address this issue by including refined details of the thermoelectric behavior of the superconducting tape [16]. These models rely on lumped parameters implementing the nonlinear resistance of the HTS tape via the power law relating the voltage along the tape to its DC current. Such model is essentially valid around the critical current ($I_{c}$) [17]. This classical approach is very efficient as it can be easily scaled to infer trends on power systems. It is particularly useful to solve both the heat balance equation and the electric equations under the same framework using nodal analysis coupling thermal and electrical physics of the superconductor. Indeed, an equivalent circuit, for both the electrical and thermal state variables, is built to resolve the temperature and voltage across and along the tape, respectively [18]. Such a model can be tuned to simulate the physics of the problem with user-defined accuracy while balancing the computational time. From the circuit point of view, the obtained voltages and currents are almost identical in both cases. It is the preferred choice for modeling complex power systems to study power grids of several nodes since it yields fair results retaining the essence of the superconductor physics [19]. This approach has proven to be very appealing to study the superconductor components in power systems.

For studying the performance of the superconductor itself, the Finite Element Method (FEM) has been the preferred choice as it includes geometrical details that lead to an accurate estimation of distributed fields such as the current density and the magnetic field over the HTS layers yielding a fair estimation of their local losses [20,21]. Such refined analysis is often sought when the quantity of the material has to be minimized or to optimize the design of the superconducting device. It should be noted that it is a numerical method relying on the spatiotemporal discretization of partial differential equations for transient analyses. By essence, it leads to an approximated solution which accuracy depends on the level of geometric details and the fineness of the mesh used to discretize the geometry [22]. Nevertheless, with the advances in computer technology with more powerful products available to the general public alongside the development of numerical computation, a new trend in the modeling of superconducting power components could emerge, combining the sophisticated and accurate FEA with circuit analysis [23]. The main advantage of using FEM-based models to simulate HTS components is the high degree of precision achieved on the local representation of the physics of the superconductor. These models can consider both the *E*-*J* relation and the critical current density dependencies on temperature and magnetic flux density (magnitude and direction). However, these models, that include different dynamics (thermal and electromagnetic), are heavy in terms of computational load, currently impairing the study of complex electrical grids involving various power items with a simulation time covering several power cycles. In our opinion, their usefulness lies in the design tuning of the superconducting component itself and not in the study of the power system as a whole, at least for the time being.

In this context, the present work summarizes the recent developments led by the authors in the modeling of 2G HTS tapes using lumped parameters and FEM coupled with external circuits. The purpose is to provide an overview of the current developments in the modelling of commercial 2G HTS tapes in the context of power systems, the current needs of the modelling community to achieve the highest accuracy to describe the dynamic behavior of superconducting power devices through up to date models and their corresponding case studies. It also discusses the impact of the variability in the superconducting properties that may affect the correct operation of those devices and the potential issues that are actually impairing their rapid penetration in the national energy grids.

After recalling the main characteristics of the REBCO technology, the models are detailed, and some examples are provided to show their respective utility. Additionally, they allow demonstrating the advantages of both approaches to tackle different scales of refinement used to study the thermoelectric behavior of REBCO in power systems. It is shown through the different examples that an accurate knowledge of $I_{c}$, *n* index, and the heat exchanged between the cryogen and the tape surface are key parameters to model a superconducting device. Even though these parameters can be measured, experimental data are scarce. No empirical scaling laws of the $I_{c}$ and *n* index, implementing the required dependencies on temperature and magnetic field over a broad range of values of interests, are available to facilitate the bridge between the applied superconductivity community and the power systems engineering community to this date.

The paper is divided into four principal sections subsequent to this introduction: (1) a review of commercial REBCO tapes supplied by some of the well-established manufacturers, (2) a presentation of the models with examples, (3) discussions, and (4) conclusions. Following a brief presentation of commercial coated conductors in the second section, the third section deals with the lumped-parameters models followed by their application in AC and DC conditions. These examples have already been published in the literature [18,24]. However, additional materials were included in this work. The fourth section presents the FEM-based model and how it may be coupled with external circuits. In contrary to the lumped-parameters models, which describe the overall thermoelectric behavior of the HTS component providing global parameters (current, voltage and power), the FEA looks at the performance of the superconductor itself, computing distributed fields (current density, magnetic field, and loss distribution). For the latter, the global parameters can be computed by integrating the distributed fields. Two examples introduce 2D and 3D FEM models for completeness. A general discussion follows in the fifth section on the current necessities of the modelling community dealing with a technology lacking standardization and very dependent on operating conditions. Finally, the work concludes by proposing some research axes to make the technology readily available for applications with a prospective large market in the electrical sector.

## 2. REBCO Material

Nowadays, there are several manufacturers of coated conductors with the capability to supply tapes in length of kilometers with low-resistance splices [25,26,27,28]. A common commercial 2G HTS tape (REBCO or RE-Ba${}_{2}$Cu${}_{3}$O${}_{7-\delta }$, RE for Rare Earth) is a multi-layer composite having the following basic layering: a top silver layer, a thin superconductor (SC) ceramic layer, a buffer layer to structure the superconductor deposit on top of a nickel alloy layer referred to as substrate and a bottom silver layer for a total thickness of less than 0.1 mm. The thickness of the layers is pretty much standard, even though it can vary slightly between manufacturers. A tape can be purchased with extra layers providing thermoelectric stabilization in the shape of electroplated or soldered high purity copper. Instead of copper, brass may also be used to reduce losses in AC conditions. A mechanical reinforcement (stainless steel) can also be added for applications requiring handling high forces. For all these possible variants, some common issues arise from the manufacturing process. Besides the splice-less length of a few hundreds of meters, engineering properties such as the $I_{c}$ and *n* index are found uneven along the tape length in addition to the anisotropy in the magnitude of the critical current according to the direction of the applied magnetic field. This anisotropy leads to the definition of a critical current that is parallel ($I_{c\|}$) and orthogonal ($I_{c⊥}$) to the tape surface [29]. Variations as much as a few percent can be found along the tape length and across batches [30]. These variations do not include additional local degradation that may arise during handling such as winding or during cooling for instance. To the best of the authors’ knowledge, such local variability has not been taken into account in past and present models. Often, the minimum or averaged $I_{c}$ is taken as the critical current of the tape or the superconducting component. In the lumped-parameters and FEM models, the inhomogeneity and anisotropy of the *n* index are not taken into account. A constant value $n_{0}$ ranging broadly from 20 to 45 is used instead [31,32]. For $I_{c}$, the inhomogeneity is often omitted. However, it is now customary in FEM models to add the anisotropy through a Kim-like relation [33,34,35]. On a side note, the critical temperature $T_{c}$ is commonly chosen between 89 K and 91 K. The following Table 1 and Table 2 summarize some of the salient characteristics of commercial REBCO tapes from catalogs.

## 3. Modeling Based on Lumped Parameters

In the following section, the basis of lumped-parameters models is laid out. The equations represent the thermoelectric behavior of a single tape, assuming uniform and homogeneous properties. These properties depend on temperature and current. To model the full transition of the HTS from its superconducting state to its normal-resistive state, the power-law is extended by simulating a parallel resistance circuit. This parallel circuit allows simulating the redistribution of the current between the superconducting film and the surrounding metallic layers. In the following section, the main characteristics of the thermoelectric model of REBCO technology are introduced. Further details can be found in the following references [18,37].

### 3.1. Extension of Power-Law Model

The analogy between thermal and electrical quantities for representing the macroscopic physics of superconductors was first presented in [38]. It was implemented to tackle the modelling of resistive Superconducting Fault-Current Limiters (r-SFCL). It has been the basis for developing a generic thermoelectric model of REBCO tapes to simulate power systems based on superconductors, including resistive or inductive components [24,39].

The power-law model is usually given as the relation between the electrical field *E* and the current density *J* of the superconductor, as shown in Figure 1. This figure illustrates the full span of electrical changes that the superconductor undergoes as the current is increased. Initially, the superconductor is in its superconducting state, and there is no electric field. Then, as the current is increased, it slowly enters a dissipating phase through different regimes, flux creep, and flux flow. The critical current density ($J_{c0}$), which represents a measure of the superconductor transport current capability, is found around the transition between creep and flux flow regimes. As the current approaches $J_{d}$, the superconductor quickly loses its superconducting state to enter its ohmic resistive state with a *n* index equal to 1. The impact of increasing temperature is also illustrated by showing how the *n* index and the critical current density $J_{c}$ tends to decrease as the temperature of the superconductor increases. More details on the underlying physics of the transition process can be found in [40,41].

As metallic layers surround the superconductor, when an electric field is induced in the HTS, part of the current is redistributed in the neighboring layers. Such behavior can be caught by obtaining experimentally the *V*-*I* characteristics of the tape. This classic measurement is performed by a four-point method using two voltage taps installed along the tape length and by feeding the tape with a known DC current through copper connections. The *V*-*I* characteristics are commonly performed at 77.3 K (in boiling liquid nitrogen, at atmospheric pressure) and in self-field (no applied external magnetic field). The *V*-*I* characteristics can be then used to infer directly the dynamic resistance of the full tape and not just the superconductor as provided by the *E*-*J* relation. The *V*-*I* characteristics is illustrated in Figure 2a. This characteristics can be easily built in a lumped-parameters model, as it relies solely on global variables such as voltages and currents to catch the essential physics of the superconductor instead of distributed fields such as the electric field *E* and the current density *J* that are more readily appropriate for FEA.

The power-law formulated in terms of the voltage *V* along the superconductor and its DC current *I* takes the following form [31],
(1)$$V=V_{c}{\left(\frac{I}{I_{c}}\right)}^{n},$$
where $V_{c}=E_{c}l_{\mathrm{tp}}$, with $E_{c}$ the critical electric field ranging from 0.1 to 10 μV/cm depending on the size of the system under characterization. $l_{\mathrm{tp}}$ is the distance between the voltage taps along the tape. This model is typically used to fit the experimental *V*-*I* characteristics to infer the $I_{c}$ and *n* index. It assumes that the electric field is homogeneous across the tape cross section at any position and any time. Such approximation is valid when the current transition length across the different layers or tapes in the superconductor device has a negligible impact on its thermoelectric and magnetic behavior. In fact, the model has a rather small range of validity around the curvature of the *V*-*I* characteristics (around $I_{c}$). This limitation can be overcome either by relating the electrical field *E* to the current density *J* covering the creep and flux flow regime of the superconductor as described in [38] or by combining the resistance of the superconductor derived from (Equation 1) with the equivalent resistance of the metallic layers in the tape. This alternative approach was proposed in [18] to solve the current sharing between the tape layers during the transition of the superconductor from its superconducting state to its normal-resistive one. Practically, it is carried out using an equivalent parallel resistance circuit model of the tape, assuming that the superconductor’s resistance in its normal state is larger than the combined resistance of the metallic layers. The resistance of the tape (superconductor plus metallic layers) is then given by,
(2)$$R_{\mathrm{tp}}=\frac{R_{\mathrm{sc}}R_{m}}{R_{\mathrm{sc}}+R_{m}},$$
with $R_{m}$ is the lumped resistance of the metallic layers depending on temperature. The resistance of the superconductor layer is obtained from the power law as,
(3)$$R_{\mathrm{sc}}=\left(\frac{V_{c}}{I_{c}^{n}}\right)I^{n-1}.$$

Here the resistance is given in Ohm. However, it can be easily normalized to the tape length with unit of Ohm per meter by using the critical electrical field $E_{c}$ in (Equation 3) instead of the critical voltage $V_{c}$.

Thus, using an equivalent parallel resistance of the tape, it is possible to extend the *V*-*I* characteristics as shown in Figure 2a. Figure 2b illustrates the corresponding model of the tape resistance. The characteristics, measured at 77 K in liquid nitrogen and self-field, provides the reference values for the critical current $I_{c0}$ and *n* index $n_{0}$ for the model. $I_{c}$ is actually the primary parameter to relate the electrical and thermal models of the conductor as well as the magnetic model if required by the application. Not widely used, further refinements can also include the temperature and magnetic field dependency of the *n* index. A general dependency may be expressed as follows,
(4)$$\left\{\begin{array}{cc} I_{c} & =I_{c0}\times I_{c,T}\left(T_{\mathrm{sc}}\right)\times I_{c,B}\left({\bar{B}}_{\mathrm{sc}}\right),\\ n & =n_{T,B}\left(T_{\mathrm{sc}},{\bar{B}}_{\mathrm{sc}},n_{0}\right), \end{array}\right.$$
where $I_{c,T/B}$ and $n_{T,B}$ are empirical functions that can be fitted to actual measurements. $T_{\mathrm{sc}}$ is the temperature of the superconductor layer, and ${\bar{B}}_{\mathrm{sc}}$ is the average magnitude of the magnetic flux density over the tape as proposed in [39].

Some power applications do not present significant magnetic field other than the self-field of each individual tape so that $I_{c}$ and *n* index dependencies can be reduced to,
(5)$$\left\{\begin{array}{cc} I_{c} & =I_{c0}\times I_{c,T}\left(T_{\mathrm{sc}}\right),\\ n & =n_{T}\left(T_{\mathrm{sc}},n_{0}\right). \end{array}\right.$$

This latter model is the one used in the subsequent examples presenting the modelling of non-inductive superconducting coils for Superconducting Fault-Current Limiters (ScFCL) [18] and Superconducting Power Filter (ScPF) [24].

### 3.2. Dependency Function $I_{c,T}$

The $I_{c}$ dependency on temperature is assumed linear around the reference temperature $T_{\mathrm{LN}2}$ as follows,
(6)$$I_{c,T}=\left\{\begin{array}{cc} 1, & T_{\mathrm{LN}2}\leq T\leq T_{\mathrm{cs}}\\ \frac{T_{\mathrm{cs}}-T}{T_{\mathrm{cs}}-T_{c}}, & T_{\mathrm{cs}}<T\leq T_{c}\\ 0, & T>T_{c} \end{array}\right.$$

The reference temperature $T_{\mathrm{LN}2}$ is the temperature of the liquid nitrogen at 77 K, for which $I_{c0}=I_{c}\left(T_{\mathrm{LN}2}\right)$.

It is a common feature of models to assume such linear behavior around 77 K which has been verified experimentally through the measurements of the critical current density as a function of temperature over a wide range of temperatures from 77 K down [36,42]. However, it is not clear that such linear model holds close to the critical temperature [43] as not enough experimental data are available beyond 77.3 K to provide a clear picture on the relation between the critical current and the temperature of the superconductor.

In this model, the current-sharing temperature $T_{\mathrm{cs}}$ is introduced. It corresponds to the temperature for which the current starts distributing between the superconductor layer and the metallic layers of the tape, as illustrated in Figure 3.

For REBCO, for which the critical current is mostly linearly dependent on temperature, it can be expressed as,
(7)$$T_{\mathrm{cs}}≃T_{c}+\left(T_{\mathrm{LN}2}-T_{c}\right)\frac{\left|i\right|}{I_{c0}},$$
with *i* the current flowing through the tape.

Simpler models may only rely on the critical temperature and the reference temperature $T_{\mathrm{LN}2}$ so that,
(8)$$I_{c,T}=\left\{\begin{array}{c} \frac{T_{c}-T}{T_{c}-T_{\mathrm{LN}2}},\;\;T_{\mathrm{LN}2}\leq T\leq T_{c}\\ 0,\;\;\;T>T_{c} \end{array}\right.$$

### 3.3. Dependency Function $n_{T}\left(T_{\mathrm{sc},n0}\right)$

Very few measurements on the dependency of the *n* index on temperature are available [29]. To include the dependence of the *n* index on the temperature, the following empirical model was employed [18],
(9)$$n_{T}=n_{0}\frac{T_{\mathrm{LN}2}}{T_{\mathrm{sc}}},$$
where $n_{0}$ is the reference *n* index at $T_{\mathrm{LN}2}$.

### 3.4. Distribution of Current: Current-Sharing Regime

When the HTS is in its superconducting state ($T_{\mathrm{sc}}<T_{\mathrm{cs}}$), the current flows entirely in the superconductor layer and $i_{\mathrm{sc}}=i$, as shown in Figure 3. When the superconductor has fully transited to its normal-resistive state ($T_{\mathrm{sc}}>T_{c}$), the current flows entirely in the metallic layers and $i_{m}=i$. However, during the transition from the superconducting state to the resistive-normal state ($T_{\mathrm{ref}}<T_{\mathrm{sc}}<T_{\mathrm{cs}}$), the current redistributes between the superconductor layer and the metallic layers. This redistribution is completely related to the temperature evolution of $I_{c}$ through the power law. Assuming a linear dependence between the critical current and the temperature of the superconductor and a homogeneous electrical field across the tape, the solution of the following polynomial provides the current flowing through the superconductor layer $i_{\mathrm{sc}}$,
(10)$$i_{\mathrm{sc}}^{n}+\frac{I_{c}^{n}}{V_{c}}R_{m}\left(i_{\mathrm{sc}}-i\right)=0.$$

The current flowing in the metallic layers is given by $i_{m}=i-i_{\mathrm{sc}}$ where *i* is the operating current either in AC or DC conditions.

This approach can be used to solve the current distribution in a stack of parallel tapes if needed. However, in most applications, the tapes are assumed identical undergoing a uniform transition over their entire length. A local transition may be considered using a propagation model as proposed in [44]. It is often an undesired situation that can lead to the local degradation or damage of the tape and hence the permanent disruption of the proper operation of the system. Such complexity can actually found its utility for quench and protection system design.

The estimation of the current distribution in the tape is based on a current divider equation as follows,
(11)$$i_{k}=\frac{R_{k}}{R},$$
with k indexing the layer in the tape, $R_{k}$ the resistance of the kth layer and *R* the tape resistance. Assuming that the time step is small enough and the temperature is known a priori, the latter formula can be directly used to avoid solving the polynomial (Equation 10) altogether as proposed in [38]. However, this approach assumes that the relevant operation of the device is achieved when the superconductor fully transits. For devices relying on the current-sharing regime, it is not the preferred choice and Equation (Equation 10) should be solved in conjunction with (Equation 11).

### 3.5. Power Dissipation

To know the distribution of temperature in the tape, it is necessary to infer the Joule dissipation in its different layers. In the layer k, the dissipated power is simply computed as,
(12)$$P_{k}=R_{k}i_{k}^{2}.$$

The heat dissipation is then used as a term source in the heat balance equation as discussed in the next section.

### 3.6. Thermal Model

Figure 4 illustrates the electrical circuit and the thermal nodal model side by side to compute the temperature distribution from the current redistribution and Joule powers in the tape layers. Each temperature node is at the center of a layer, and each node is connected to the next via thermal resistances. This model is coupled to the electrical one through the dissipated power given by (Equation 12) providing a heat source to each node. The field of temperature is obtained by solving a set of heat balance equations lumped into the following expression,
(13)$$T^{t+\Delta t}={\left(\frac{1}{\Delta t}C^{t}+W^{t}\right)}^{-1}\cdot \left(\frac{1}{\Delta t}C^{t}T^{t}+W_{\mathrm{bd}}^{t}T_{\mathrm{LN}2}+P^{t}\right),$$
where W is the matrix of thermal conductances in [W/K] derived element by element from the inverse of the thermal resistance *R*^th^ connecting one node to the next. The thermal resistance is given by $R^{\mathrm{th}}=\frac{th_{\mathrm{tp}}}{k_{k}A_{k}}$ where $th_{\mathrm{tp}}$ is the thickness of the layer k, $A_{k}$ is its cross-sectional area, and $k_{k}$ is the equivalent thermal conductivity across adjacent layers. $W_{\mathrm{bd}}$ is the boundary convective thermal conductance representing the cooling between the top and bottom layers of the tape and the nitrogen bath. The convective thermal conductance is obtained by linearizing the relation between the heat flux *q* exchanged at the boundary and the temperature difference ΔT between the node at the boundary of the solid and the LN2 so that $q=W_{\mathrm{bd}}A\Delta T$, with *A* the exchange surface. C is the matrix of thermal capacitances. It is a diagonal matrix made of the heat capacitances of the diverse materials associated with their respective temperature node accounting for the increase in internal energy during transient dissipation (not shown in Figure 4). The heat capacitance in [J/K] is obtained from the specific heat capacity $c_{p},k$ for each layer k as $C_{k}=\gamma_{k}V_{k}c_{p,k}$ where $\gamma_{k}$ and $V_{k}$ are the mass density and the volume of the layer k, respectively. The thermal conductivites, heat capacities and the electrical resistivities $\rho_{k}$ build in the resistances of each layers ($R_{k}=\rho_{k}\frac{l_{\mathrm{tp}}}{A_{k}}$, $l_{\mathrm{tp}}$ is here the length of the tape) depend on temperature according to the data provided by the National Institute of Standards and Technology (NIST) [45]. P is the power dissipated in the tape layers as provided by (Equation 12), and *t* and Δt represent the current time and the time step, respectively. More details of the thermal model including the definition of each term of (Equation 13) can be found in [37].

### 3.7. Examples of Model Usage in AC and DC Applications

The following section presents two case studies using the model described above. The first case study is an application of r-SFCL in a three-phase AC network. The second case study deals with a ScPF as part of an aircraft DC grid. In both cases, the devices are built on a non-inductive superconducting coil referred to as superconducting component (SCC) operated in liquid nitrogen at 77 K. The SCC relies on a versatile design that can be modelled under the same framework as presented previously. Even though the conceptual design of the SCC is similar for both case studies, their operation differs. The r-SFCL relies on the rapid transition of the superconductor to its normal-resistive state while the ScPF is exclusively operated in the current-sharing regime.

In terms of modelling, some adjustments have to be made, since in both cases a shunt resistor is added in parallel to the SCC. Additionally, the r-SFCL utilizes several tapes connected in parallel and co-wound together to manage the current magnitude exchanged with the power grid. To accommodate these extra features, the Equation (Equation 10) is modified as follows,
(14)$$i_{\mathrm{sc}}^{n}+\frac{I_{c}^{n}}{V_{c}}R_{m,\mathrm{sh}}\left(i_{\mathrm{sc}}-N_{\mathrm{tp}}i\right)=0,$$
where,
(15)$$R_{m,\mathrm{sh}}=\frac{R_{m}R_{\mathrm{sh}}}{R_{m}+N_{\mathrm{tp}}R_{\mathrm{sh}}}.$$

In (Equation 14), $R_{\mathrm{sh}}$ is the resistance of the shunt and $N_{\mathrm{tp}}$ is the number of parallel tapes per turn. For this new equation, all the tapes are assumed to be the same, all fully cooled by LN2, and therefore, behave identically. Under these assumptions, only the current redistribution of a single tape has to be solved to obtain the thermoelectric response of the SCC to transients. In this case, the current does not only divide between all the metallic layers but also with the shunt resistor as shown in Figure 5. Therefore, to model the redistribution of current, Equation (Equation 11) must be modified as well by replacing *R* with the equivalent resistance of the device (tapes plus shunt). Comprehensive details on the case studies with their respective models and simulations with additional analyses can be found in the following references [18,24].

#### 3.7.1. Conceptual Design of the Superconducting Component (SCC)

The SCC is constituted of series-connected coils made of commercial 2G HTS tapes operated in liquid nitrogen at 77 K. Between the turns in each coil, an insulator guarantees electrical insulation while still providing maximum cooling as proposed in [46]. For a non-inductive configuration, the winding of a superconducting coil is such that its inductive component is canceled by winding back the tape on itself [18,47]. An additional metallic shunt resistor, which can be either located at room temperature or co-wound with the tape [48], serves as a bypass to the current during transients. This additional shunt can provide supplementary control on the amount of resistance reached by the device besides enabling passive protection to the superconducting coils during a full transition from the superconducting to the normal-resistive state of the superconductor.

#### 3.7.2. Resistive Superconducting Fault-Current Limiter (r-SFCL)

Figure 6 shows a three-phase circuit comprised of a 1 GVA conventional synchronous generator (or a lumped model of several generators) connected to the AC power grid modelled as an infinite bus at 60 Hz via a transformer and a 700 km transmission line. At the terminals of the machine, three identical r-SFCL are installed, one per phase. In steady state, the generator supplies 950 MW at a power factor of 0.95.

The r-SFCL is made of a non-inductive SCC with a parallel shunt resistor located at room temperature (see Figure 5). To get the proper current margin in nominal operation, the coil is wound with a bundle of 225 parallel tapes. The r-SFCL was modelled using the temperature dependency of the critical current given by (Equation 8). It is the most appropriate model to simulate the thermoelectric response of the SCC when it is to be operated in its normal-resistive state. This operation is chosen to provide a quick response to a transient fault with current limitation and power compensation for stabilization. Table 3 summarizes the details of the case study and the main characteristics of the power components [18].

A three-phase fault is applied at the transmission lines for 100 ms (5 power cycles). The simulation time is 20 s (1200 power cycles) for which the total computation time including the post-processing lasts 1500 s, a little longer than the simulation time. Figure 7 provides the results of the first 600 power cycles for the cooling curve CC-1 given in Figure 8. Figure 7a shows the time evolution of the rotation velocity of the machine rotor normalized to its nominal velocity $\omega_{0}$ for three cases: (1) No fault, (2) fault without r-SFCL, (3) fault with r-SFCL. For the first case, the rotation remains unchanged at its nominal value showing that the basic model responds as expected. However, if a fault occurs and no protection or limitation is provided through r-SFCL, as in case 2, the rotor speed increases before diverging quickly. By adding a r-SFCL per phase, as in case 3, it is possible to limit effectively the impact of the fault and stabilized the machine so that, once the fault clears, the machine recovers its nominal regime. In this case, the r-SFCL consumes enough active power keeping the balance between the constant mechanical power provided by the machine turbine and the electrical power exchanged with the circuit. Figure 7b shows the evolution of the RMS current in phase “a” coming from the machine stator. In addition to providing stability, the r-SFCL limits the fault current, which is its key feature, preventing possible damages to the equipment installed in the circuit line. Figure 7c provides an overview of the current-sharing process occurring during the transition of the SCC and the recovery process as described by (Equation 14). When the superconductor has undergone a full transition to the normal-resistive state, the current redistribution is given by (Equation 11).

Using this case study, a series of parametric investigations were performed to evaluate the impact of the cooling and the choice of the *n* index on the results. Figure 8 shows three different LN2 cooling curves extracted from literature [49,50,51]. The differences between the curves are found in the magnitude of the maximum peak cooling power, the onset of the convection cooling, and the onset of the film boiling regime. Figure 9 summarizes the impact of the choice of cooling curves on the evolution of the superconducting layer temperature and the resistance of the r-SFCL obtained from the model. The dynamic response of the r-SFCL resistance is unequivocally determined by the time evolution of the temperature as shown in Figure 9a,b. The worst case scenario is the adiabatic case for which there is no cooling available to recover the superconducting state. The resistance remains at the value of the shunt resistance with a steady increase of the SCC temperature. When cooling is available, recovery is possible and the temperature may stabilize depending on the characteristics of the cooling curve as discussed hereinafter. As shown in Figure 8, the curves CC-1 and CC-2 have very similar features, such as a similar temperature difference at the onset of convection cooling (1 K) and film boiling (30 K). Such similar features lead to the same thermal response of the SCC, leading to the same resistance evolution over time. The thermal response with curve CC-3 is however different with a quicker recovery of the superconducting state (recovery time $t_{r}$). The curve CC-3 shows clear distinct features from curves CC-1 and CC-2. The temperature difference at the onset of convection cooling is about 7 K (instead of 1 K for CC-1 and CC-2), the magnitude of the maximum peak cooling power is notably lesser than the one of curves CC-1 and CC-2 (at least a factor of 3 lesser), and the film boiling regime is triggered at a temperature difference around 50 K (instead of 30 K). The two former parameters have a negligible impact on the thermal response as seen in Figure 9a. However, the last feature, the onset of film boiling regime, appears to be the singular parameter to accelerate the recovery. In this case, the transition from film boiling to convection cooling occurs at a larger temperature (50 K instead of about 30 K). Therefore, as the dissipated power extraction improves at a larger temperature, the superconductor can be cooled down below its critical temperature sooner thereby accelerating the recovery of the superconducting state.

The magnitude of the reference index $n_{0}$ was varied to derive its impact on the thermoelectric response of the r-SFCL. In this parametric study shown in Figure 9, the cooling curve CC-1 was used. The arrows indicate the time evolution. Discrepancies in the temperature (Figure 10a) and resistance (Figure 10b) are related to the initial onset of the transition of the superconductor to its normal-resistive state. The impact is larger for the resistance as the process follows closely the current evolution through the power-law model in the current-sharing regime (corresponding to $I_{a,\mathrm{RMS}}$ between 40 and 100 kA). The choice of the *n* index has no impact on the thermoelectric response of the r-SFCL when the superconductor is fully transited ($R_{r-\mathrm{SFCL},\mathrm{RMS}}≃0.08\;\Omega$) or during the recovery process. At the onset of transition, the impact of the index is felt. At the lowest values of $n_{0}$ (→15), the transition is smoother with increasing current. In contrast, at the largest values (→35), the transition is quicker with a temperature reaching sooner the critical temperature of the superconductor ($T_{c}=91\;K$). For a reference value above 35, the discrepancies are miscellaneous, and the choice of the reference *n* index may be made based on numerical convenience by helping the convergence of the solver.

#### 3.7.3. Superconducting Power Filter (ScPF) for Embedded DC Grids

The previous parametric analyses showed that the choice of the cooling curve as well as the reference *n* index matters if a refined study should be carried out in the current-sharing regime. For the ScPF operated in this regime, the worst cooling scenario, provided by the curve CC-1 with the smallest temperature difference at the onset of film boiling, was initially chosen. A somewhat low but still in the range of reference *n* indexes equal to 20 was also chosen.

Figure 11 shows the ScPF built on top of a classic RLC filter. The SCC is similar to the one found in the r-SFCL presented in the previous section, with a shunt resistor connected in parallel [52]. In the following case study, an ideal DC power supply (ps) provides a controlled power load (cpl). Table 4 presents the parameters of the case study and the ScPF. More details on the case study and the applications can be found in [24,53].

Figure 12 shows the power drawn by the load during transients. The basic stability specifications of the system and some information about the operation of the ScPF are presented in Figure 13. Thus, the voltage oscillations should be less than 0.1% of the mean voltage (${\bar{V}}_{\mathrm{CPL}}$) at the load within the time of the power surge ($\Delta t_{p}$), and the maximum temperature of the superconductor ($T_{\mathrm{sc},\max}$) cannot pass its critical temperature at any time.

One of the key aspects of the ScPF design is the minimization of conductor length to achieve stability under the conditions set in Figure 13. The minimum tape length is partly responsible for the overall cost of the device as well as its weight. In the subsequent study, the minimum tape length $l_{\mathrm{tp},\min}$ for a 3 mm wide tape at a minimum critical current $I_{c0}$ from the manufacturer’s catalog (SuperPower Inc.) equal to 75 A is analyzed. Figure 14 shows the resulting voltage and current for two cases concerning a 54% gain in power limit stability compared to the stability limit of the pure RLC filter ($P_{\mathrm{RLC}}=145.8$ kW). The first case is the SCC operated in its current-sharing regime with a minimum tape length of 0.72 m. The second case corresponds to the SCC undergoing a full transition to the normal-resistive state for a tape length equal to 0.672 m. Figure 15 summarizes the thermoelectric response of the ScPF. In the current-sharing regime, the temperature never passes the superconductor’s critical temperature and the recovery of the initial state is quasi immediate, as shown in Figure 15a. The resistance of the SCC is strictly described by the power-law following closely the current behavior. The resistance initially oscillates before stabilizing, as shown in Figure 15b. If the superconductor fully transits to the normal-resistive state, there is not enough cooling to provide thermal equilibrium, and the temperature increases until the power surge clears to reach a maximum value of 263 K. The recovery of the initial state takes about 3 s. The resistance initially follows the current oscillations since the superconductor is still in its current-sharing regime. Then, as the HTS is no longer in the superconducting state, the resistance follows the evolution of the temperature instead of the current. At this point, the current has fully redistributed in the shunt and metallic layers, having a lower equivalent resistance than the resistance of the superconductor in the normal-resistive state. In the current-sharing regime, the current redistributes between the superconductor layer, the metallic layers, and the shunt as shown in Figure 15c.

Figure 16 shows the negligible impact of the reference index $n_{0}$ on the minimum tape length for the cooling curve CC-1. At a low end reference *n* index ($n_{0}\to 15$), the minimum tape length increases by 6%. For $n_{0}>25$, the minimum tape length does not depend on the reference *n* index. In the case of the impact of different cooling conditions provided by the cooling curves given in Figure 8, the peak heat flux is actually the relevant parameter to determine the operating conditions of the ScPF. Table 5 shows the minimum tape length obtained at $n_{0}=20$ for the different cooling curves. As the cooling degrades following a lower peak heat flux, the minimum tape length has to be drastically increased to achieve stability under the stability criteria given by Figure 13. No tape length could be found for CC-3 in the present case. It is therefore sensible to get the best achievable cooling conditions to ensure the well and safe operation of the ScPF.

In terms of computation time, a run for a 20 s simulation was carried out for consistency with the computation time of the r-SFCL simulation. In the present case study, the computation time is 532 s compared to the 1500 s for the r-SFCL case. The computation time is a lot quicker than the time of the simulation.

## 4. Modeling Based on Finite Element Method

In this section, a different approach using FEM is taken to model the HTS device with a primary focus on the performance of the superconductor material itself. For such approach, all the electric conductive metallic layers are fused into the surroundings. The surroundings are modelled as “air” (or non-conductive domain). However, it should be noted that, when using the *H* formulation of the Maxwell equations and edge elements, a fictitious electrical resistivity is also associated with the “air” [54]. It is implicitly assumed that the equivalent resistivity of the metallic layers is a lot larger than the resistivity of the superconductor. It is a valid assumption as long as the superconductor is operated below or around its critical current density.

In the present work, the FEM model is coupled to an external circuit and the SCC is then part of an electrical grid modelled as lumped parameters [23]. The recent *T*-*A* formulation of the Maxwell equations is used to describe the electromagnetic behavior of the superconductor [55,56,57,58,59]. For this new formulation, a new method for coupling the superconducting device and the electrical grid considering all the non-linearities present in the system (ferromagnetic and superconductor) has been purposely developed. The initial model was two dimensional (2D) [23] but it was quickly expanded to three dimensions (3D) [60]. These models are presented hereinafter. Subsequently, 2D and 3D examples of a simple superconducting coil connected to an external circuit are detailed. In the 2D model, the power losses are computed via two methods for self consistency [23,34,61,62].

### 4.1. T-A Formulation

The *T*-*A* formulation solves two variables, the Current Vector Potential (CVP) T and the Magnetic Vector Potential (MVP) A. Figure 17 presents the main concept associated with the formulation. In the MVP domain ($\Omega_{A}$), the source is the current density which is divided into three parts: (1) the external current density ($J_{e}$), (2) the induced current density ($J_{i}$) that are computed in the MVP domain, and (3) the superconductor current density ($J_{\mathrm{HTS}}$) that is calculated in the CVP domain ($\Omega_{\mathrm{HTS}}$). Using the MVP, the magnetic field is computed in the whole domain knowing the *B*-*H* relation for the material through B=∇×A, whereas the CVP is used to calculate the current density only in the superconducting regions [55,63,64,65]. Equations (Equation 16) and (Equation 17) present the *T*-*A* formulation.
(16)$$\nabla \times \left(\rho \nabla \times T\right)=-\frac{\partial B}{\partial t},$$
(17)$$\nabla \times \left(\frac{1}{\mu }\nabla \times A\right)=J_{\mathrm{HTS}}+J_{e}+J_{i},$$
where $J_{\mathrm{HTS}}$ is the current density flowing through the superconductor layer ($J_{\mathrm{HTS}}=\nabla \times T$), **B** is the magnetic flux density, ρ is the electrical resistivity, and μ is the magnetic permeability ($\mu_{0}$ for the superconductor and the “air”). The external electrical circuit is coupled to the FEM model via $J_{e}$. This current density is computed from the current flowing through the circuit model.

The thin current sheet approximation assumes that the current flows tangentially to the superconducting layer [66]. Therefore, the current can only follows the longitudinal direction of the thin conductor [67]. In 3D, the tape reduces to a 2D sheet as shown in Figure 18. In 2D, the thin conductor is approximated by a line, and the current density is restricted to flow in only one direction.

Using the thin current sheet approximation, the **T** and **A** formulations are coupled by a continuity condition,
(18)$$n\times \left(H_{1}-H_{2}\right)=d_{\mathrm{HTS}}\;J_{z},$$
where **n** is the unit vector normal to the tape surface, $H_{1}$ and $H_{2}$ are the magnetic field vectors above and below the tape, respectively. The total transport current *I* is imposed by:(19)$$\left(T_{z2}-T_{z1}\right)d_{\mathrm{HTS}}=I.$$
with $d_{\mathrm{HTS}}$ the thickness of the HTS layer.

As in the case of the lumped-parameters models, the power-law model is also used to represent the physics of the HTS material. However, the electric field *E* is then related to the current density *J*. From this relation, the electric resistivity (ρ) is inferred as,
(20)$$\rho \left(J_{\mathrm{HTS}},B\right)=\frac{E_{c}}{J_{c}\left(J_{\mathrm{HTS}},B\right)}{\left|\frac{J_{\mathrm{HTS}}}{J_{c}\left(J_{\mathrm{HTS}},B\right)}\right|}^{n-1},$$
where $E_{c}$ is the critical electric field, ranging from 0.1 to 10 μV/cm depending if it is a short sample or a coil; *n* is the *n* index. $J_{c}$ is the critical current density, which depends on the temperature and on the magnitude and direction of the magnetic flux density.

For the FEM model, only the superconductor electromagnetic behavior is modelled, and the temperature of the tape is assumed constant at 77 K. However, the magnetic flux density dependence of $J_{c}$ is taken into account via an elliptical relation given in (Equation 21). It is an important contrast with the lumped-parameters models that used temperature dependence as their key feature. It gives, for the lumped-parameters models, a larger range of operating conditions that the FEM model has thus far provided. As mentioned previously, the FEM relies only on the power-law model assuming an operation around the critical current density.
(21)$$J_{c}\left(B_{⊥},B_{//}\right)=\frac{J_{c0}}{{\left(1+\sqrt[{}]{\frac{\gamma^{2}B_{//}^{2}+B_{⊥}^{2}}{B_{0}^{2}}}\right)}^{\alpha }},$$
$B_{0}$ is a parameter expressing the $J_{c}$ rate of decay with the magnitude of the magnetic flux density, α is a damping parameter, γ is the anisotropy factor and $J_{c0}$ is the critical current density in the absence of any external magnetic field at 77 K (Self-Field).

### 4.2. Coupling FEM and External Electrical Circuit

Various authors have already carried out the coupling between FEM and external circuit for conventional systems [68,69,70]. These systems rely on typical conductors such as Cu and/or Al to transmit the current assuming constant material properties. For such coupling, the *A*-*V* formulation has been successfully employed. It is the *de facto* formulation to model conventional technologies. Due to the highly non-linearity of the *E*-*J* relation in a superconductor, different formulations leading to better convergence were tried out. Amongst the most successful formulations, the *H* formulation relying on the magnetic field as its state variable has been widely used. Lately, a new *T*-*A* formulation has proved to be more efficient in terms of computation speed and memory usage for modelling 2G HTS equipment [9,35,58,71,72,73,74,75]. However, it is still too complex and too expensive computationally to simulate a whole HTS device with all its elements. Therefore, only the superconductor layer in a coated conductor is typically modelled for large-scale devices. To mitigate this issue, lumped parameters can be used to represent some part of the device. The FEM model is used to simulate the HTS layer alone, thereby reducing drastically the computation time. The advantage of the coupling method is related to the ability to study the superconducting device in a power system while retaining fine details of the HTS electromagnetic behavior.

The coupling method proposed in [23] and applied in [6] provides a proper way to consider all the material non-linearities for both the superconductor and the ferromagnetic core (when present) while still allowing an easy coupling with electrical lumped-parameters systems in 2D and 3D. Through the FEM, the electrical parameters of the superconducting device are calculated and passed to the power grid model. The circuit computes global values such as current and voltage. The current is then fed back to the FEM model as an engineering current density source $J_{e}$. Figure 19 summarizes the whole method. On the left side, the superconducting device is modelled using FEM. All the remaining power components or power systems that are not built on the superconductor are then modelled as lumped parameters such as circuit breakers, power electronics, power plants, etc. These equipment or systems are shown on the right side of Figure 19. The coupling method facilitates the communication between these two different models (distributed field with FEM, and global variables with the circuit). An iterative procedure was implemented to integrate both models. First, the electrical parameters are computed via FEM, assuming initial conditions on the current source. Using the finite difference method, the electric circuit is then solved, and a new current is devised. This current is fed anew to the superconducting device until convergence on the current is achieved. The way the current is impressed in the FEM model depends on the formulation. For the *T*-*A* formulation, a Dirichlet condition is used as given in (Equation 19).

For the circuit analysis, the superconducting device is represented by a voltage $V_{\mathrm{HTS}}$ as shown schematically in Figure 20. This voltage is obtained from the integration of the electric field computed by the FEM. The electric field is obtained from,
(22)$$E=-\nabla V-\frac{\partial A}{\partial t},$$
where *V* is the electric scalar potential. By integrating and reformulating Equation (Equation 22), the voltage along the superconducting tape is computed as follows,
(23)$$V_{\mathrm{HTS}}=\int_{l_{\mathrm{tp}}}\nabla V\cdot dl=-\int_{l_{\mathrm{tp}}}E\cdot dl-\int_{l_{\mathrm{tp}}}\frac{\partial A}{\partial t}\cdot dl,$$
where $l_{\mathrm{tp}}$ represents the path along the tape length.

In the next section, the proposed methodology is applied to simulate examples in 2D and 3D.

### 4.3. Examples of the Use of the Coupling Method

For the 2D and 3D examples, a stabilizer-free 12 mm wide commercial REBCO tape is modelled with a minimum critical current of 300 A and a superconducting layer thickness of 1 μm.

#### 4.3.1. 2D FEM-Circuit Coupling

This section details the implementation of the coupling method in 2D in a simple inductive superconducting coil. If multiple turns are considered per coil, and various coils are modelled, the voltage accumulated along all the turns and coils connected in series is given by,
(24)$$V_{\mathrm{HTS}}=\sum_{k=1}^{k=N_{\mathrm{st}}N_{l}}V_{\mathrm{HTS},k},$$
where $N_{\mathrm{st}}$ and $N_{l}$ are the number of coils and turns per coil, respectively. $V_{\mathrm{HTS},k}$ is the voltage per turn at turn k computed from (Equation 23) as,
(25)$$V_{\mathrm{HTS},k}=-\left(\left[\frac{\int_{w}E_{z}+dy}{\int_{w}dy}-\frac{\int_{w}E_{z}-dy}{\int_{w}dy}\right]+\frac{d}{dt}\left[\frac{\int_{w}A_{z}+dy}{\int_{w}dy}-\frac{\int_{w}A_{z}-dy}{\int_{w}dy}\right]\right)l_{\mathrm{tp},k},$$
where $E_{z}$, and $A_{z}$ are the electric field and the magnetic vector potential in the *z* direction, orthogonal to the *x*-*y* plane given in Figure 21, and $l_{\mathrm{tp},k}$ is the tape length at turn k. The integration calculates a weighted average, where the path *w* of the integration follows the width of the tape. The geometric parameters of the 2D case, illustrated in Figure 21, are given in Table 6. The coil is coupled to an AC current source. The current has an amplitude of 150 A (50% of $I_{c0}$) and a frequency of 50 Hz.

Figure 22 shows the magnetic flux density and the normalized current density at 15 ms. Figure 23 compares the current calculated by the circuit, which uses a finite difference method for solving the problem, and the current calculated by FEM whose total current is calculated by (Equation 26) as,
(26)$$I=\left(\int_{w}J_{z}dy\right)\;t_{\mathrm{HTS}},$$
where $t_{\mathrm{HTS}}$ and $J_{z}$ are the thickness of the superconducting layer and the current density in the *z*-direction, respectively. As it can be observed in Figure 23, both currents agree, indicating that the coupling algorithm between the circuit and the FEM model has successfully converged.

The coupling method allows to split the powers arising from the resistive and inductive currents using (Equation 22). The power associated with the resistive current in the superconducting device is the active power or the losses given by,
(27)$$P_{a}=\left(\sum_{i=1}^{N_{\mathrm{st}}N_{l}}\int_{w}P_{a,\mathrm{el}}dy\right)\;t_{\mathrm{HTS}},$$
where,
(28)$$P_{a,\mathrm{el}}=\left(E_{z,\mathrm{el}}\;l_{\mathrm{tp},\mathrm{el}}\right)\;J_{z}.$$
$E_{z,\mathrm{el}}$ is the electric field orthogonal to each element of the superconducting layer mesh, and $l_{\mathrm{tp},\mathrm{el}}$ is the 2G tape length corresponding to the position of the element in the mesh.

Figure 24 shows a comparison between the active power calculated by both the traditional method and the coupling one. There is a very good agreement between the results.

From (Equation 22), it is possible to extract the reactive power arising from the electric field induced by the time evolution of the magnetic field as follows,
(29)$$P_{r}=\left(\sum_{i=1}^{N_{\mathrm{st}}N_{l}}\int P_{r,\mathrm{el}}dy\right)\;t_{\mathrm{HTS}},$$
where,
(30)$$P_{r,\mathrm{el}}=\left(-\frac{\partial A_{z,\mathrm{el}}}{\partial t}\right)\;l_{\mathrm{tp},\mathrm{el}}\;J_{z}.$$
Figure 25 shows the evolution of the reactive power. As predicted by the electrical circuit theory, this power does not produce any work with a RMS value equal to zero. It corresponds to the power stored in the magnetic field.

As previously done for the examples of the lumped-parameters models, the first analysis using the coupled FEM-circuit model looks at the impact of the reference *n* index $n_{0}$ on the losses. For the FEM model, the *n* index found in the power law does not depend on the temperature and the magnetic field, it is equal to the constant value $n_{0}$ which is a common assumption for such models. The same range of values is used here as well (15, 20, 25, 30, and 35). A second analysis looks at the impact of the critical current $I_{c0}$ on the losses for values spanning ±10% of the minimum critical current provided by the manufacturer. In the present case, the minimum critical current is equal to 300 A for a 12 mm wide tape.

Figure 26 presents the results for both analyses. The losses increase with the increase in the *n* index (Figure 26a), whereas the losses decrease as the reference $I_{c0}$ increases (Figure 26b). As the primary goal is always to reduce the losses, a low *n* index (→15) similar in magnitude to those found in BSCCO tapes (first generation or 1G HTS wire) would be desirable with a large $I_{c0}$. The impact of $I_{c0}$ being larger than that of the *n* index, it is the primary parameter to reduce effectively the losses.

An additional study was carried out on the computation time and the sensitivity of the active power (losses) on the fineness of the tape mesh. Figure 27 presents the impact of the choice of number of nodes discretizing the tape width. The relative error on the RMS active power was estimated over the second power cycle using a reference value $P_{r,\mathrm{ref}}$ of 120 nodes. Additionally, the computation time was estimated for each number of nodes. The relative error on the RMS active power is given by,
(31)$$er_{P}=100\times \left(\frac{P_{r,n}-P_{r,\mathrm{ref}}}{P_{r,\mathrm{ref}}}\right)$$
where $P_{r,n}$ is the computed RMS active power for $N_{\mathrm{nd}}$ nodes in the tape. As expected, more mesh elements lead to a larger computation time. There is a saturation in the relative error therefore on the accuracy of the result as more elements are added to the tape. For the present studies, the reference number of nodes of 120 was used to discretize the tape for a computation time equal to 558 s. The computation time, encompassing the combined FEM and circuit computation times, evolves linearly with the number of nodes per tape.

#### 4.3.2. 3D FEM-Circuit Coupling

This section presents the 3D FEM case study corresponding to the simulation of a HTS single phase power transformer. This simple example may illustrate the potential of the proposed methodology for actual electric power equipment. Fundamentally, it could be used to represent precisely the physics of the following items in a global model at the same time: conventional conductor, superconductor, ferromagnetic material, and external electric circuit components.

On the low voltage side of the transformer, it uses a superconducting coil wound with 6 turns of insulated coated conductor. On the high voltage side, a conventional coil is wound with 30 turns of insulated copper wire. For this example, the coils are wound on a core made of ferromagnetic material. This core allows guiding the magnetic flux from the primary to the secondary. Both non-linearities due to the superconductor (in the *T* formulation) and the ferromagnetic material (in the *A* formulation) are included in the model. Using the proposed coupling method, both short-circuit and open-circuit classic tests are simulated. The FEM mesh with 45,241 elements is presented in Figure 28a. The magnetization curve of the ferromagnetic material used to simulate the core is given in Figure 28b.

Figure 29 shows the schematic drawings of the electric circuit for the short-circuit and open-circuit tests. For the short-circuit test (Figure 29a), the low voltage side (LS) having a superconducting coil is coupled with a current source symbolizing the electrical grid. On the high side (HS), the copper coil is short-circuited by a resistance *R* = 10 nΩ. For the open-circuit test (Figure 33b), the copper coil is not connected. For both tests, a current source supplies a peak value of 150 A in the superconducting coil at a frequency of 50 Hz.

The results of the short-circuit test are presented in Figure 30. The induced current in the high side (copper coil) follows the ideal transformer equation,
(32)$$\frac{i_{\mathrm{HS}}}{i_{\mathrm{LS}}}=\frac{N_{\mathrm{LS}}}{N_{\mathrm{HS}}},$$
where $i_{\mathrm{HS}/\mathrm{LS}}$ are the currents on the high side (HS) and the low side (LS), and $N_{\mathrm{HS}/\mathrm{LS}}$ is the number of the turns on the high side and the low side, respectively.

Figure 31 shows the magnetic flux density and the normalized current density for the first half of power cycle. The core is not saturated in this example. The distribution of current density is uneven across the tape width as expected in AC conditions when the current is lower than the critical current. When the magnetic flux changes, an induced voltage appears across the terminals of the coils as electromotive forces are created according to the Faraday-law of induction. In steady-state regime, a mutual magnetic flux flows through the magnetized iron core. Figure 32 shows the waves in the low and high voltage sides. The same ideal transformer relation was checked as well using,
(33)$$\frac{V_{\mathrm{HS}}}{V_{\mathrm{LS}}}=\frac{N_{\mathrm{HS}}}{N_{\mathrm{LS}}}.$$

The voltage at the low side is the product between the low-side current and the impedance.

For both the short-circuit and open-circuit case scenarios, the relative error and computation time were estimated as a function of number of nodes in the tape. Figure 33 shows the results. The tape was divided in 150 elements along its length for all the simulations. As the number of nodes increases, the relative error for a reference RMS active power of 24 nodes decreases. The dependency is linear. However, the computation time does not increase linearly which is mainly due to convergence issues at very low numbers of nodes. For 24 nodes across the width of the tape, the computation time is about 18 h for the short-circuit case and about 39 h for the open-circuit case.

## 5. Discussion

The previous studies showed the utility of lumped-parameters models and coupled FEM-circuit models with different power-law refinements. For the lumped-parameters models, the dependence of $I_{c}$ and *n* index on temperature and/or magnetic flux density (if the application requires [39]) is employed for which there is not enough experimental data to derive empirical law for the *n* index dependencies and to some extent for the critical current, particularly above 77.3 K. These models allow simulating the behavior of the full superconducting device as part of a power system. For simulation runs of several power cycles considering various parallel-connected tapes, the computation time extends to about 25 min for the most demanding r-SFCL case study. For the coupled FEM-circuit model, the critical current is assumed to depend on the magnetic field and its orientation, with no temperature dependency. For such models, the current flowing in the HTS device is expected to remain around $I_{c}$, and no transition of the superconductor from its superconducting state to its normal-resistive state is ever simulated limiting the scope of the FEM-circuit coupling model. The heat dissipation is assumed negligible, and the temperature remains at the operating temperature. For this coupled model, the computation time for the 2D case is of the order of hundreds of seconds depending on the targeted accuracy (>100 s). For the 3D case, it is larger with a computation time that covers several hours (12 nodes for 6 h in the 2D case and 8 nodes in the 3D cases with 3 h in the short-circuit case and with 8 h for the open-circuit case). Both models are complementary and find their respective utility. The lumped-parameters models offer a wider range of possible superconducting device operations in complicated electrical networks with a relatively fast computation time. In contrast, the coupled FEM-circuit models provide deep insight into the device’s operation and the performance of the tapes near their critical current in simple electric circuits with a larger computation burden than that of the lumped-parameters models. Both models were simulated on similar machines. The lumped-parameters models were run on a 3.1 GHz Quad-Core Intel Core i7 processor with 16 GB - 2133 MHz LPDDR3. The FEM-circuit models were run on an AMD Ryzen^™^ 7 2700 eight-core processor with a 16 GB at 3.2 GHz.

From the examples, it appears that the main characteristics of commercial 2G HTS tapes for their application in power systems are the critical current $I_{c}$ and the *n* index. The former is the primary parameter to justify the use of superconductors compared to more conventional technology since the current density is a lot larger than the one found in high electrically-conductive conductors at negligible losses. Thus, for the same power, the HTS devices get compacter and lighter. The reductions in weight and volume are particularly appealing for embedded systems on-board ships and aircrafts or for wind power for which a turbine sits on top of a large tower. In this case, the structure of the tower drives the price of the system, so having a light turbine reduces the amount of structural support and enables additional features such as offshore floating towers [76]. As shown in the examples of the lumped-parameters models, besides targeting the largest possible $I_{c}0$ at the lowest dependencies, it is also important to have a choice between different values of critical currents as well as different tape features to fit the requirements of the application. These features are the choice of stabilizer and tape width. Both features are relevant parameters to provide thermal balance with the cryogen as shown by the impact of the cooling curves. Indeed, by providing a bypass to the current when the superconductor is in its normal-resistive state, the stabilizer can sufficiently diminish the amount of power dissipation for the LN2 to be able to cool the tape. Additionally, a large tape surface can help to achieve such feature and compensate for the variation of cooling power as shown by the large discrepancies in cooling power offered by the three cooling curves (CC-1, CC-2, and CC-3). In terms of tape length manufacturing, even though the scaling of the manufacturing of splice-less long tape lengths is desired, it does not appear to be an actual issue. As a matter of fact, some large-scale applications have already demonstrated the reliability of using kilometers of tapes relying on metal-to-metal electrical connection or splices [77]. However, the main issue is the variability of the critical current along the length of the conductor (inhomogeneity) and its anisotropy concerning the direction of the magnetic field [78,79]. The manufacturers provide some statistics on the critical current along the tape length, which has not been exploited yet. For example, if $I_{c}$ measurements along the entire coated conductor are not available, the manufacturers still provide the minimum critical current. Ideally, it is required to measure the overall critical current of the SCC to have a clear set of values if such experimental capability is available. When such measurements are not available, empirical dependency functions can still rely on reference values ($I_{c}0$ and $n_{0}$) provided by the manufacturer, from third-party measurements, or the literature. Statistics are often missing from the manufacturers to be able to build a database of reference values. As all the models are built upon the power law that uses both $I_{c}$ and the *n* index, these two parameters should be readily available as well as their dependencies on temperature, magnetic field, and mechanical strain. The latter is actually relevant for application involving a large magnetic field or the possible mechanical degradation of the tape during winding [80]. Such case is often dealt with by introducing degradation parameters such as for superconducting power cables [81]. Overall, seldom data are available for REBCO tapes even though some efforts have already been done to address this issue [82]. Data are available for the dependency of the critical current on the magnetic field and/or its orientation for 4 mm wide tapes. There is, to this date, no comprehensive database to the best of our knowledge for different REBCO technologies in different operating conditions. For the *n* index, the dependence on temperature and/or magnetic field is insufficient to infer any accurate scaling laws, and only trends have been used thus far.

For the development of superconductor power systems, international standards are essential. These standards have to be defined yet so that the electrical industries can confidently employ the technology in existing power grids. Efforts in that direction have been taken for both the coated conductor basic characterization in LN2 (at 77 K) not yet for device design and operation [83]. For basic tape characterization, the idea is to provide reliable $I_{c}$ values across manufacturers measured at 77 K in LN2 through standards procedures. To achieve consistency, the measurement setup has to be clearly defined. On the device side, it is more complicated as the technology is still improving. Thus, this effort is likely to be firstly carried out for BSCCO technology as the technology has matured enough to provide a certified product. The REBCO technology is still impaired by the inhomogeneity in the superconducting properties along the conductor length.

## 6. Conclusions

The present work showed the latest developments in the modelling of HTS devices for large-scale power system applications. Two approaches were introduced, the lumped parameters equivalent models and the Finite Element Method with circuit coupling. Both approaches target distinct analyses of the superconducting device through different levels of details or scales. The lumped-parameters models provide insights into device behavior. Likewise, to the price of a greater computation effort, the FEA gives more details of the superconductor performance itself, which can be used to achieve optimal equipment design. Following the presentation of the theory, examples were given showing the thermoelectric response of REBCO tapes as well as the whole superconductor device. These examples served for advertising the need for a broad characterization campaign of commercial tapes across all manufacturers to extract the critical current and *n* index in different operating conditions. A database is presently needed for modelers to simulate the behavior and performance of HTS devices. It could provide the possibility to generate scaling laws useful for the modeler and device designer. Parametric studies were carried out to demonstrate the importance of knowing some key parameters such as the critical current and *n* index to support the soundness of the models. These parameters are built inside the power law, which is the basis of a large portion of the existing models. The knowledge of the cooling conditions is also a crucial parameter; however, it is a very complicated task to get relevant models or experimental data since the ability to extract power from the tape depends on many uncontrolled variables (surface quality, spacing, available volume of liquid, etc.). In the general discussion, some important issues to be still addressed were summarized, focusing on the requirements to be fulfilled for the superconducting devices to penetrate the electrical market. Thus, more data on the dependence of the critical current and *n* index are required to provide a clear overview of the variations in characteristics for the different commercial tapes across all the manufacturers. The limited production of the splice-less length of conductor, as yet to be resolved, does not appear to be an issue affecting the performance of superconducting device. Currently, the main issue is the lack of homogeneity of the superconductor properties along the conductor’s length, especially when a large amount of tape has to be used. The inhomogeneity is also across batches and manufacturers. The issue may be a bottleneck to ensure that there is consistency in the response between devices as well as to optimize the device for a given application during the design phase. All these issues are impairments for developing generic models of superconducting power devices to be included in power grid simulators. These simulators are essential tools in power systems engineering and provide a prediction for the grid operators on the fly and offline as well as the ability to infer planning and operations ahead of time.

## Figures and Tables

**Figure 1 materials-14-01892-f001:**
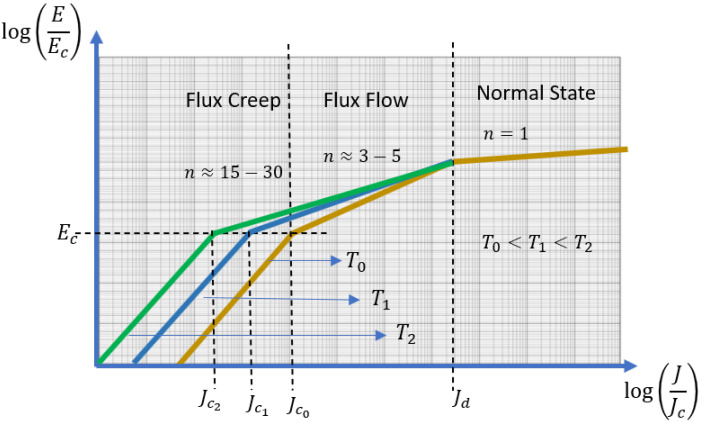
Evolution of the electric field *E* across the superconductor as a function of its current density *J*. The power-law model represents the transition from the flux creep regime to the flux flow regime. The critical current density $J_{c}$ is defined at the transition between those two regimes.

**Figure 2 materials-14-01892-f002:**
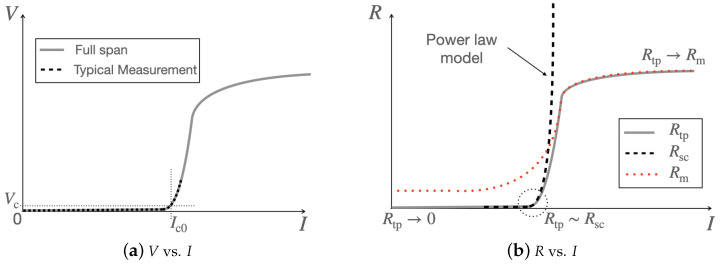
(**a**) Example of *V*-*I* characteristics. (**b**) Corresponding model of the tape resistance and extension of the power-law model in DC conditions.

**Figure 3 materials-14-01892-f003:**
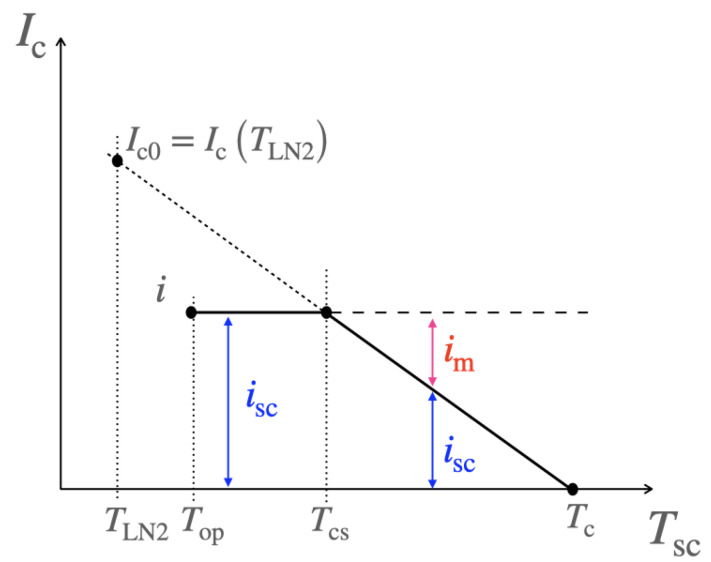
Current-sharing model. Below the current-sharing temperature $T_{\mathrm{cs}}$, all the current *i* flows through the superconductor layer. Between $T_{\mathrm{cs}}$ and $T_{c}$, the current redistributes between the metallic and superconductor layers. Above $T_{c}$, the current flows through the metallic layers.

**Figure 4 materials-14-01892-f004:**
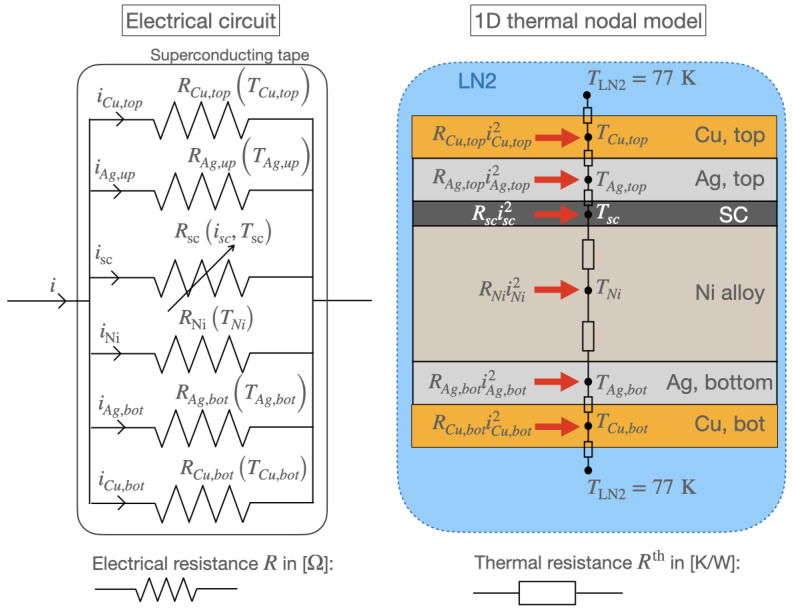
Non insulated Cu-stabilized tape. **Left**, electrical parallel circuit of a tape. **Right**, circuit model for the determination of the temperature distribution in the tape. Not shown here, each temperature node is associated with its corresponding heat capacity (see [37] for further details).

**Figure 5 materials-14-01892-f005:**
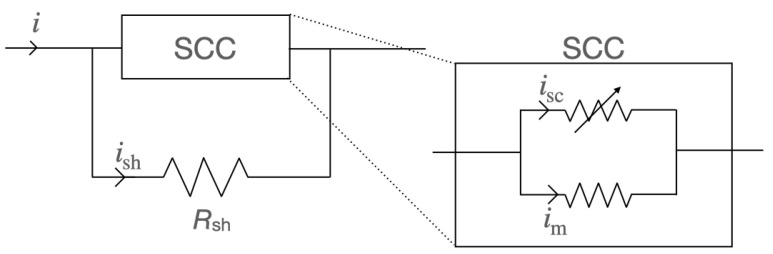
Model of a power device including a superconducting component (SCC) made of series-connected 2G HTS coils and a shunt resistor ($R_{\mathrm{sh}}$) connected in parallel to provide a current bypass.

**Figure 6 materials-14-01892-f006:**
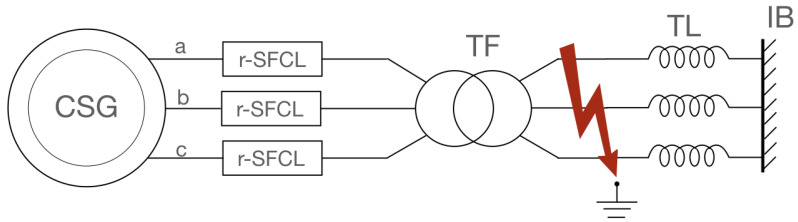
Improvement of transient stability of large conventional synchronous generators using r-SFCL technology [18]. The generator was modelled in the dq0 frame of reference.

**Figure 7 materials-14-01892-f007:**
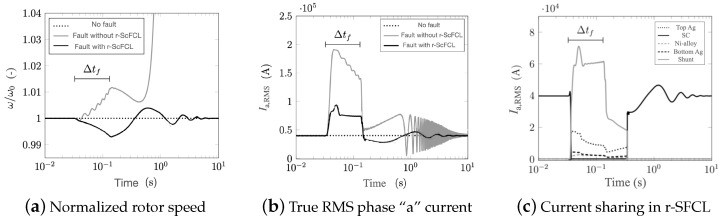
No fault and fault conditions. During the fault, the machine is stable with r-SFCL and unstable without r-SFCL. $\Delta t_{f}$ represents the duration of the fault. (**a**) Normalized rotor speed, (**b**) true RMS phase “a” current, and (**c**) current redistribution of the phase “a” r-SFCL.

**Figure 8 materials-14-01892-f008:**
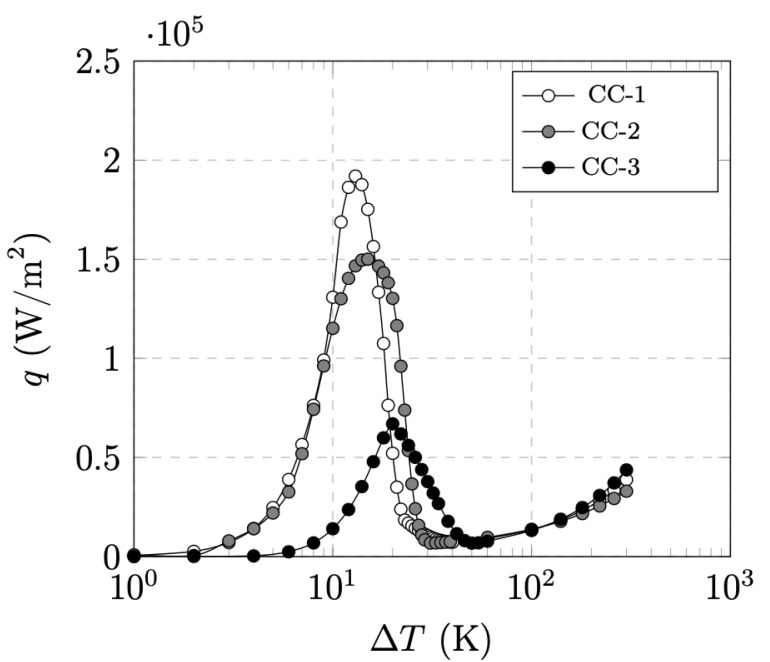
Heat flux *q* as a function of temperature difference ΔT between a LN2 bath at 77 K and a Cu surface from three different bibliographic sources. The curves CC-1, CC-2 and CC-3 are from references [49,50,51], respectively.

**Figure 9 materials-14-01892-f009:**
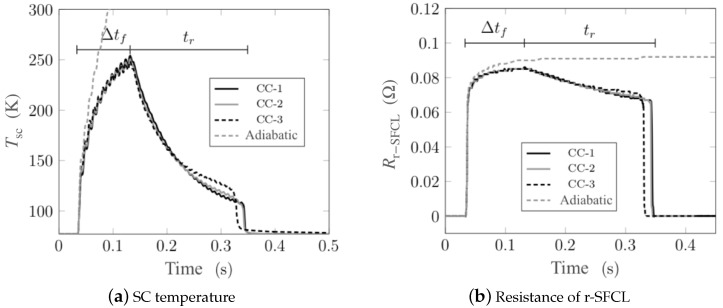
Temperature of the superconductor (**a**) and resistance of the r-SFCL (**b**) versus time for the cooling curves, CC-1, CC-2 and CC-3 in addition to the adiabatic case. $t_{r}$ is the recovery time defined as the time difference between the time at which the fault clears and the time at which the superconductor recovers its initial superconducting state.

**Figure 10 materials-14-01892-f010:**
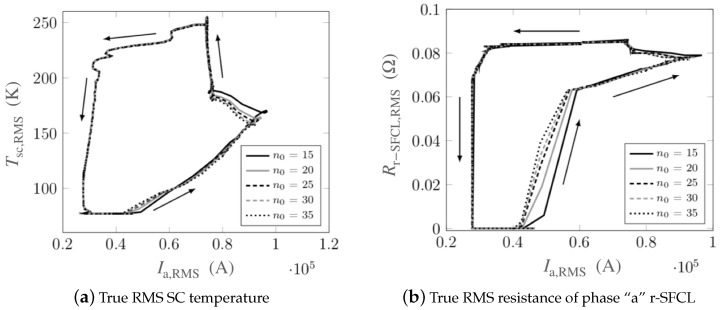
True RMS temperature (**a**) and resistance of the phase “a” r-SFCL (**b**) as a function of the true RMS phase “a” current for the reference index $n_{0}$ given by (Equation 9).

**Figure 11 materials-14-01892-f011:**
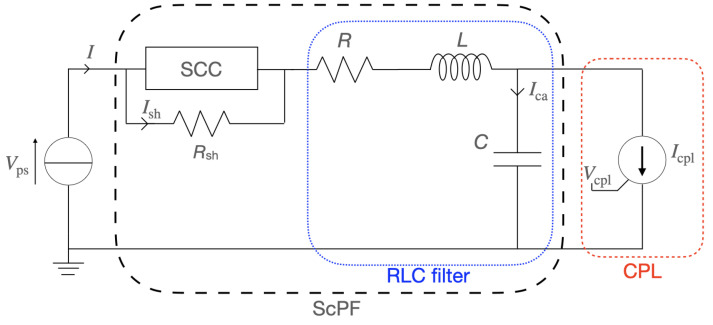
Improvement of the power stability limit of DC grids [24].

**Figure 12 materials-14-01892-f012:**
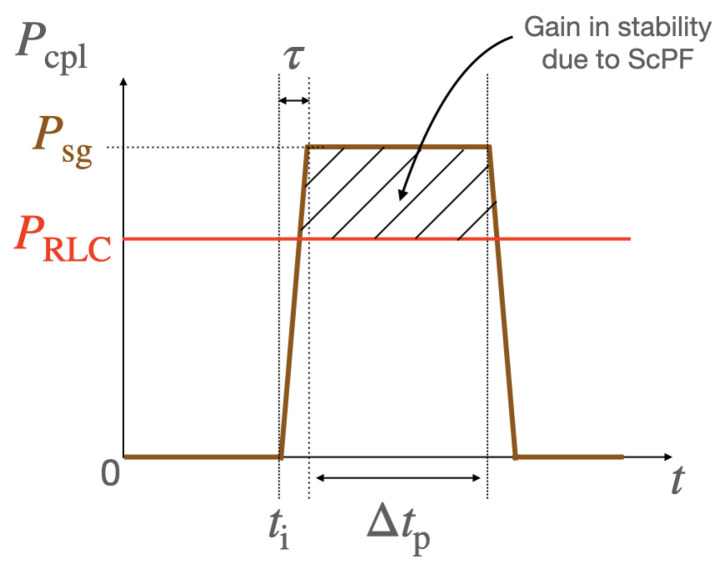
Evolution of the magnitude of the power surge $P_{\mathrm{sg}}$ over time drawn by the load (τ = 0.1 s, $\Delta t_{p}$ = 0.7 s).

**Figure 13 materials-14-01892-f013:**
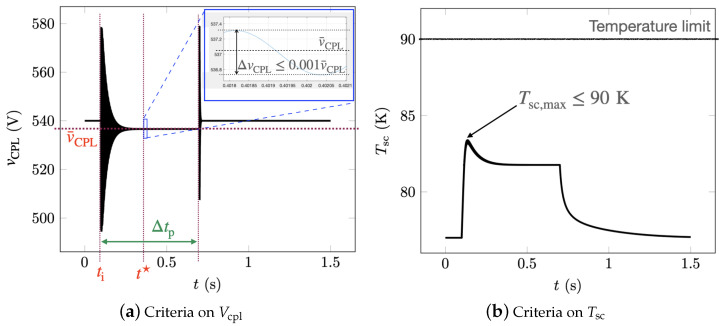
Example of the response of the load voltage $V_{\mathrm{cpl}}$ (**a**) and the temperature of the SC layer $T_{\mathrm{sc}}$ (**b**) to the power surge and definition of the stability criteria for designing the ScPF. $t^{🟉}$ is the time at which the voltage reaches less than 0.1% of the average voltage ($t_{i}-t^{🟉}<\Delta t_{p}$).

**Figure 14 materials-14-01892-f014:**
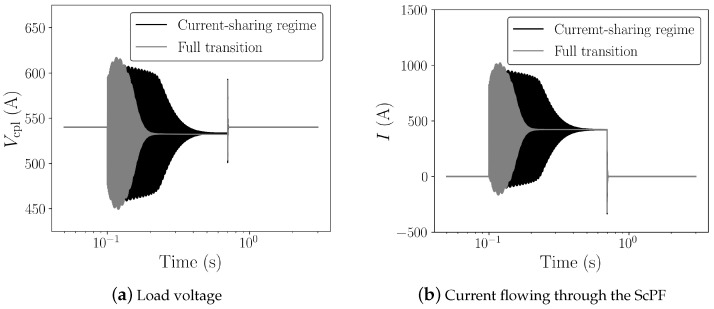
Load voltage (**a**) and current (**b**) flowing through the ScPF. The stability is largely improved when the superconductor fully transits however its temperature increases pass the critical temperature leading to possible damages to the device lowering its reliability.

**Figure 15 materials-14-01892-f015:**
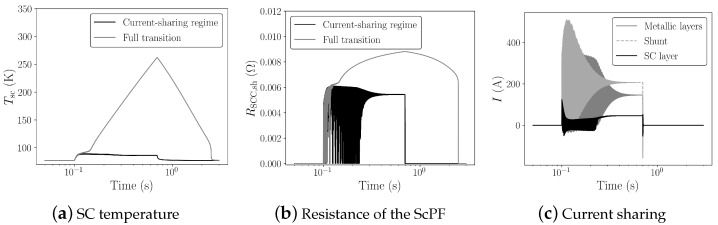
Temperature of the the SC layer (**a**) and resistance of the ScPF (**b**) for different conditions of operation of the ScPF in the DC link. (**c**) The current distributes between the SC layer, the metallic layers and the shunt in the current-sharing regime.

**Figure 16 materials-14-01892-f016:**
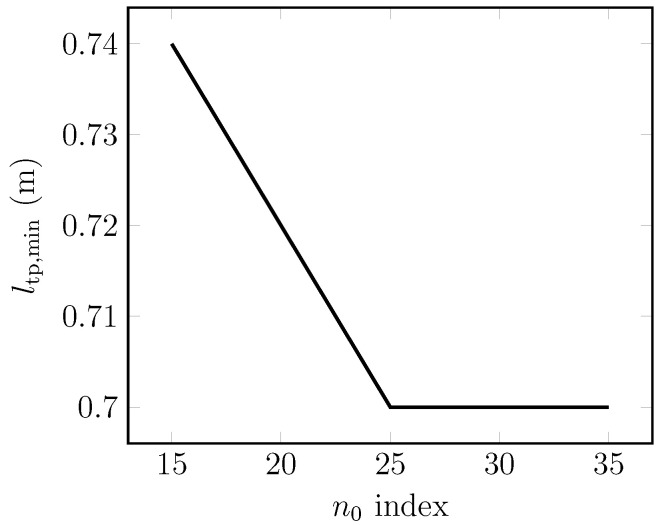
Impact of the reference *n* index $n_{0}$ on the minimum tape length to ensure stability under the design criteria.

**Figure 17 materials-14-01892-f017:**
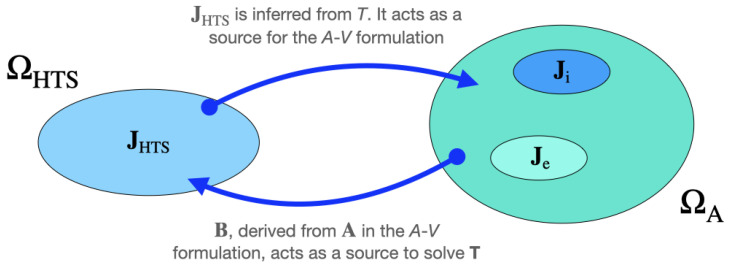
Domains associated with the *T*-*A* formulation and localization of the respective current densities given in (Equation 17).

**Figure 18 materials-14-01892-f018:**
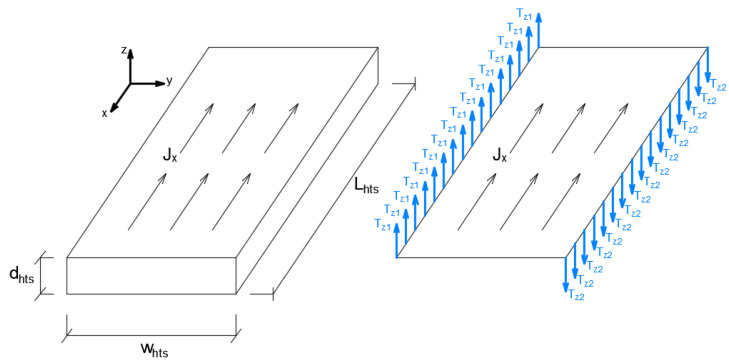
Thin sheet approximation of the current in the superconducting tape. Potentials $T_{z1}$ and $T_{z2}$ are the current vector potentials at the superconducting tape edges.

**Figure 19 materials-14-01892-f019:**
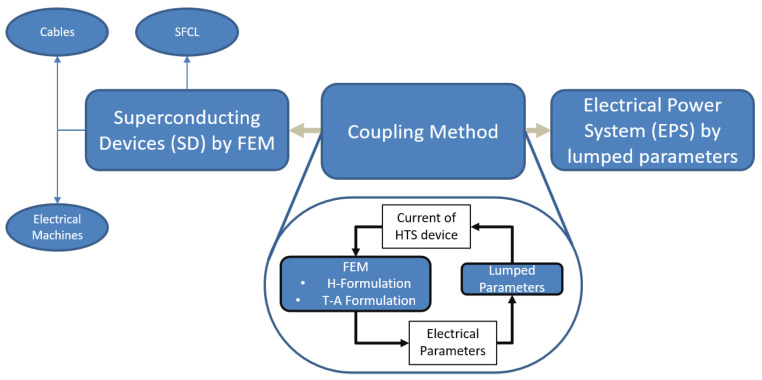
Illustration of the method coupling the 3D FEM model to the electrical circuit built with lumped parameters.

**Figure 20 materials-14-01892-f020:**
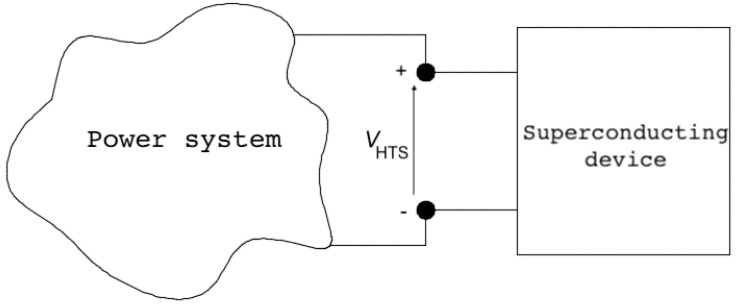
The voltage across the superconducting device computed by FEM is used in the circuit analysis referred to as the power system.

**Figure 21 materials-14-01892-f021:**
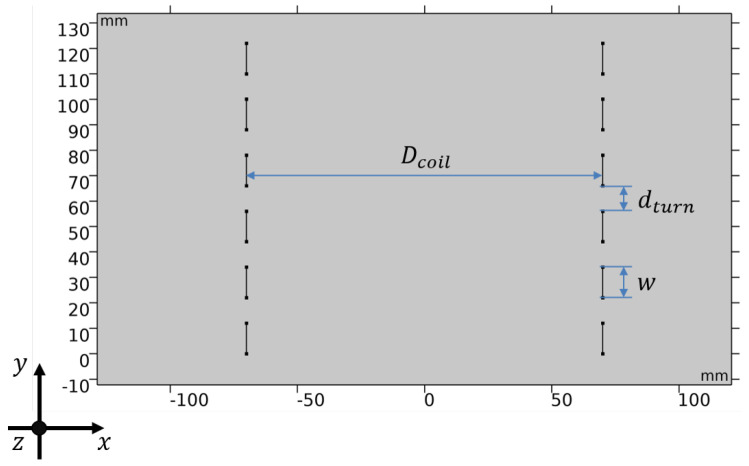
Dimensions of the coil in the 2D *x*-*y* plane and definition of the geometrical parameters (see Table 6).

**Figure 22 materials-14-01892-f022:**
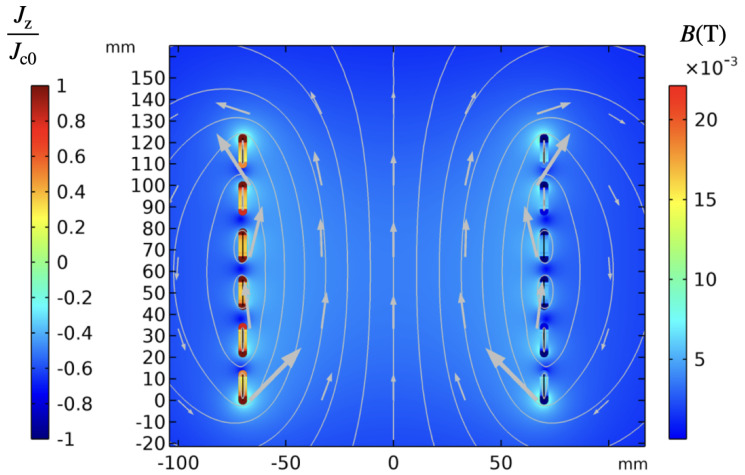
Distribution of the normalized current density $J_{z}/J_{c0}$ and map of the magnetic flux density *B*. The vectors indicate the local direction of the magnetic field.

**Figure 23 materials-14-01892-f023:**
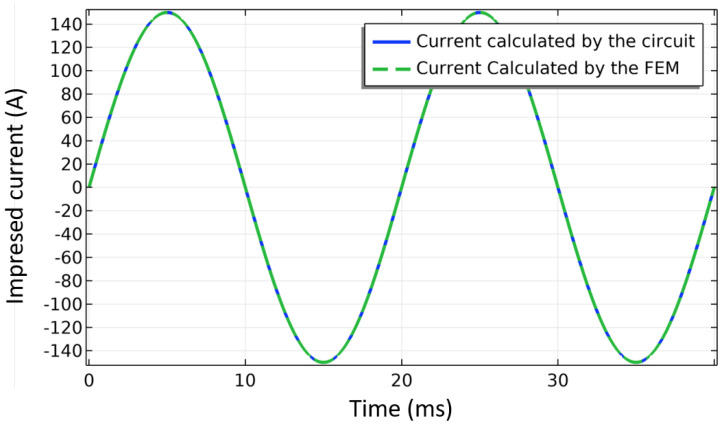
Cross-check of the computed currents in the circuit model and the FEM model. Both currents agree very well between each other indicating the correct coupling of the models.

**Figure 24 materials-14-01892-f024:**
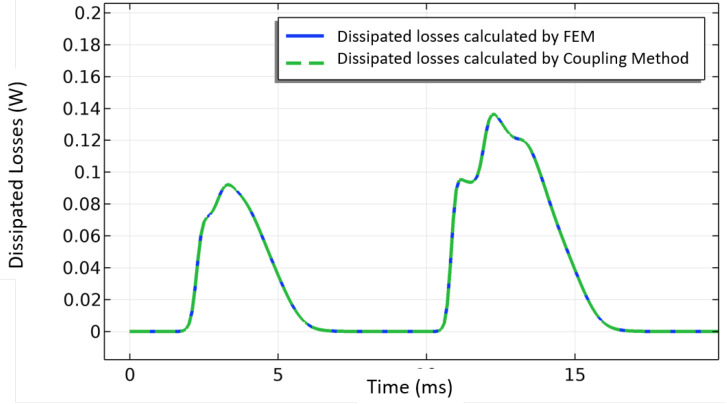
Comparison between the active power $P_{a}$ calculated via the coupling method and via the FEM.

**Figure 25 materials-14-01892-f025:**
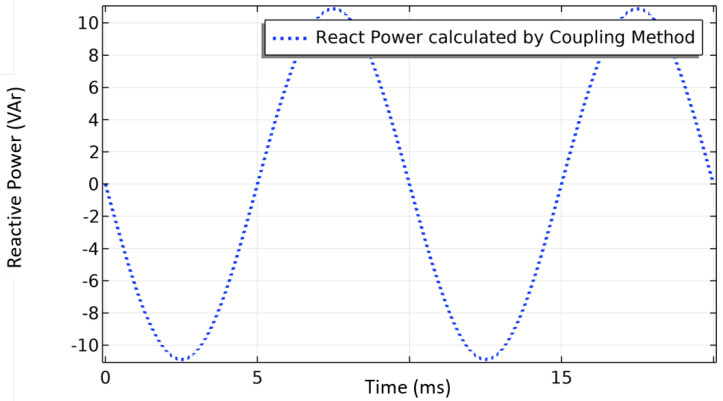
Reactive power $P_{r}$ calculated via the coupling method.

**Figure 26 materials-14-01892-f026:**
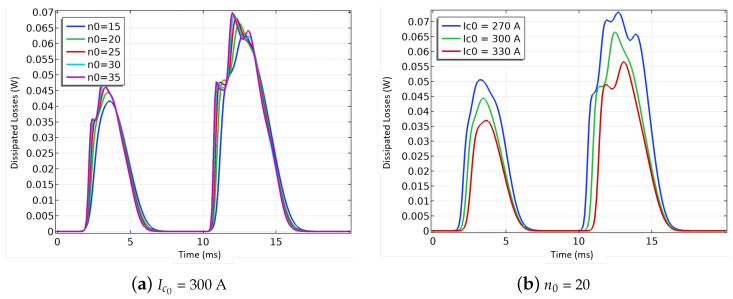
Power losses of the superconducting coil due to variations in the reference *n* index $n_{0}$ (**a**) and the minimum critical current $I_{c0}$ (**b**).

**Figure 27 materials-14-01892-f027:**
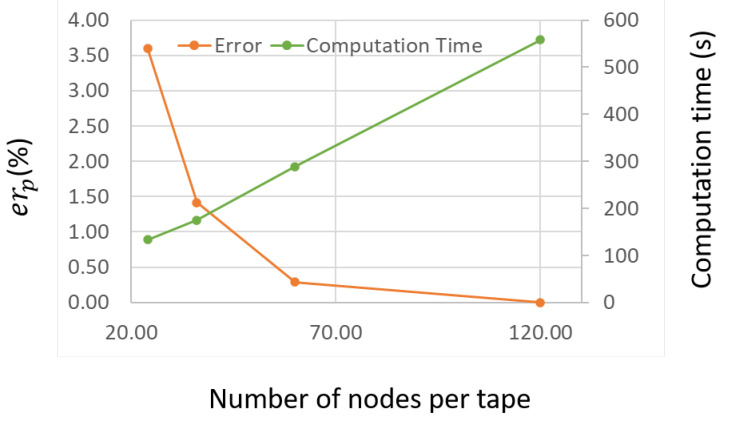
Relative errors on the RMS active power computed for different numbers of nodes discretizing the tape (orange line). Computation time versus the number of nodes per tape (green line).

**Figure 28 materials-14-01892-f028:**
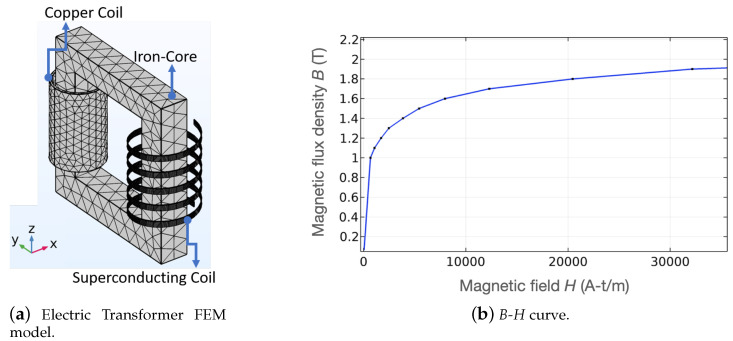
(**a**) 3D FEM model of HTS single phase transformer; (**b**) *B*-*H* curve used to model the core.

**Figure 29 materials-14-01892-f029:**
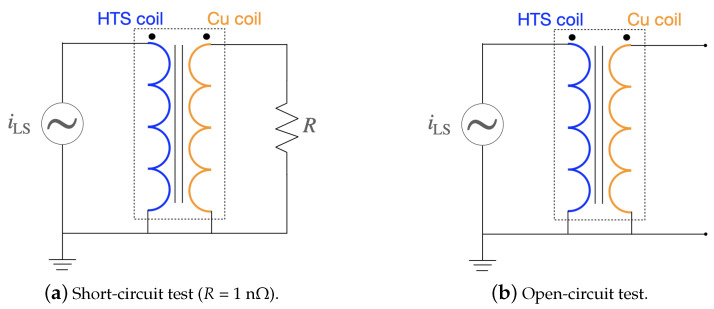
Schematic drawings of the short-circuit (**a**) and open-circuit (**b**) tests.

**Figure 30 materials-14-01892-f030:**
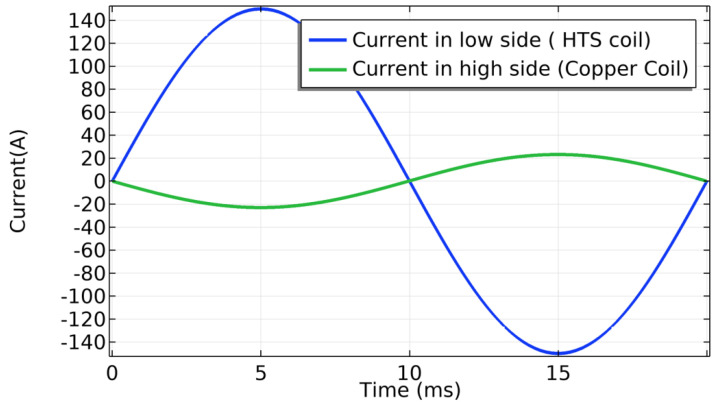
Current flowing in the high side and low side of the transformer during the short-circuit test.

**Figure 31 materials-14-01892-f031:**
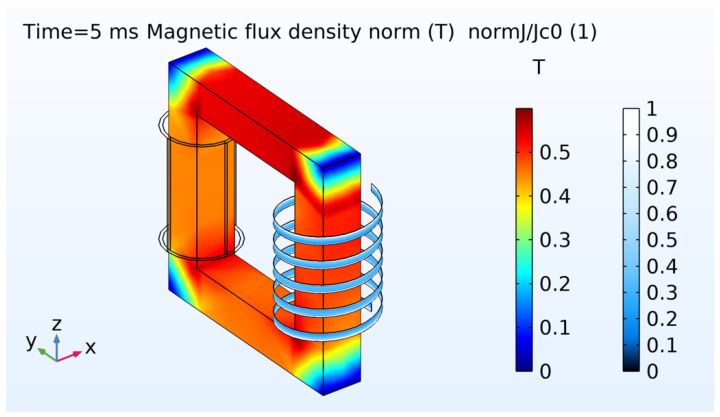
Magnetic flux density and normalized current density in the open-circuit test.

**Figure 32 materials-14-01892-f032:**
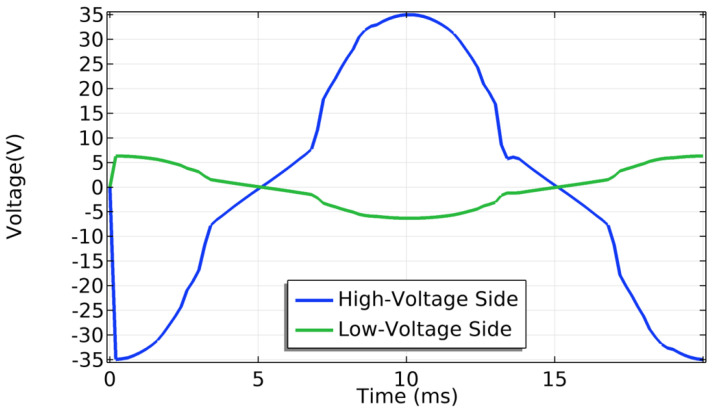
High-side and low-side voltages obtained for the open-circuit test.

**Figure 33 materials-14-01892-f033:**
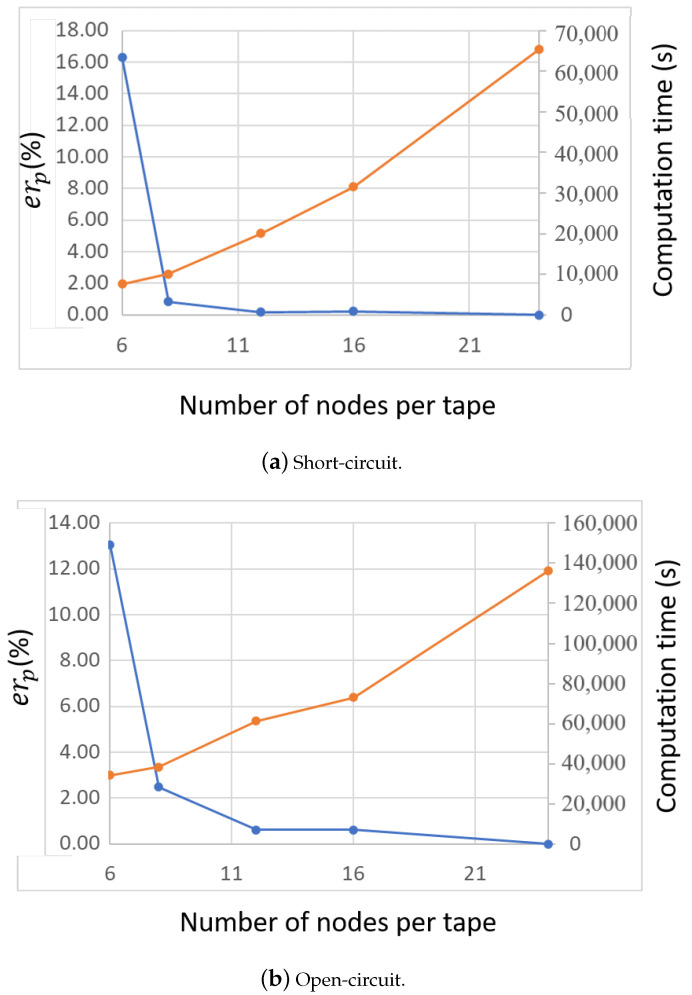
Relative error on the RMS power losses (active power) for different numbers of nodes (blue line) and the computation time (orange line) for the short-circuit (**a**) and open-circuit (**b**) cases.

**Table 1 materials-14-01892-t001:** Salient characteristics of commercial rare-earth barium copper oxide (REBCO) tapes for some manufacturers [25,26,27,28,36].

Parameter	SuperPower Inc.	SUNAM Ltd.	SuperOx	AMSC
Technology	IBASD/MOCVD	IBAD/RCE	IBAD/PLD	RABiTS/MOD
Substrate	Hastelloy	Hastelloy	Hastelloy	NiW
Cu stabilizer	electroplated	electroplated	electroplated	laminated
REBCO thickness (μm)	1.4	1.4	1.2	0.8

**Table 2 materials-14-01892-t002:** Minimum critical current in self-field ($I_{c0}$) in ampere at 77 K from catalogs [25,26,27,28].

Tape Width (mm)	SuperPower Inc.	SUNAM Ltd.	SuperOx	AMSC
2	50	-	-	-
3	75	-	-	-
4	100	100	100	80 ^1^
6	150	-	150	-
12	300	400	300	250

${}^{1}$ 4.8 mm wide tape.

**Table 3 materials-14-01892-t003:** Parameters of the electrical circuit at 60 Hz and its power components [18].

Parameter	Value	Unit
Synchronous generator		
RMS line-to-line machine voltage	13.8	kV
Inertial constant, *H*	3.7	s
Number of pairs of poles, *p*	32	-
Stator resistance	0.0029	p.u.
Reactances along the d axis, $X_{d}$, $X_{d}^{\prime }$, $X_{d}^{\prime \prime }$	1.31, 0.3, 0.25	p.u.
Reactances along the q axis, $X_{q}$, $X_{q}^{\prime }$, $X_{q}^{\prime \prime }$	0.47, 0.24, 0.18	p.u.
Time constants, $T_{d}$, $T_{d}^{\prime }$, $T_{q0}^{\prime \prime }$	1.01, 0.05, 0.1	s
r-ScFCL		
Stabilizer free tape (SuperPower Inc., Glenville, NY, USA)	SF12100	-
Tape width	12	mm
Critical current $I_{c0}$ (at 77 K, Self-Field)	300	A
Operating temperature	77	K
Number of parallel tapes	225	-
Total length of the tape	607.5	m
Shunt resistance, $R_{\mathrm{sh}}$	0.103	Ω
Delta-wye transformer		
RMS voltage (low side/high side)	13.8/500	kV
Low side impedance	0.002	p.u.
High side impedance	0.002+j0.12	p.u.
Impedance of magnetization	500+j500	p.u.
Line transmission		
Positive zero-sequence resistance	0.05, 0.605	p.u.
Positive zero-sequence inductance	0.923, 3.399	p.u.
Positive zero-sequence capacitance	1.139, 1.826	p.u.

**Table 4 materials-14-01892-t004:** Parameters of the DC electrical circuit and its components [24].

Parameter	Value	Unit
RLC filter		
Resistance of the RLC filter, *R*	0.001	Ω
Inductance of the RLC filter, *L*	10	μH
Baseline power limit stability, $P_{\mathrm{RLC}}$	144	kW
SCC		
Cu-stabilized tape (SuperPower Inc., Glenville, NY, USA)	SCS3050	-
Tape width	3	mm
Critical current $I_{c0}$ (at 77 K, Self-Field)	75	A
Operating temperature	77	K
Shunt resistor, $R_{\mathrm{sh}}$	0.01	Ω

**Table 5 materials-14-01892-t005:** Impact of the cooling on the minimum tape length.

	CC-1	CC-2	CC-3
$l_{\mathrm{tp},\min}$ (m)	0.72	1.75	-

**Table 6 materials-14-01892-t006:** Geometric parameters of the 2D case (see Figure 21).

Parameter	Value	Unit
Coil Diameter, $D_{coil}$	140	mm
superconducting tape width, *w*	12	mm
Distance between turns, $d_{turn}$	10	mm

## Data Availability

Data are contained within the article.

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
