# Peer review of "Essential Material Knowledge and Recent Model Developments for REBCO-Coated Conductors in Electric Power Systems"

_materials, 2021, doi:10.3390/ma14081892_

Round 1

Reviewer 1 Report

In this manuscript, the authors are focusing on some of the latest developments in the modeling of commercial REBCO tapes in power systems with a particular focus on the thermoelectric behavior of the superconducting device connected to external circuits.  The authors are review two important methods for the analysis and give a broad variety of examples to show their practical applications in electrical power systems. The approach of authors is making an interesting article for the reader of superconducting electronics and related scientific community. Therefore, I recommend the paper for publication

Author Response

Dear reviewer,

We would like to thank you for supporting the present work.

Best regards,

The authors

Reviewer 2 Report

This manuscript reports the models of high-temperature superconducting tapes. The authors report the modeling using lumped parameters and FEM. After the detailed explanations of the two models, the authors showed examples of thermoelectric response of high-temperature superconducting tapes and the whole device. The simulation of high-temperature superconducting tapes in a large-scale power system is important for the practical application of high-temperature superconducting materials. The lack of homogeneity of the superconducting properties along the tape length is a crucial issue. This manuscript would contribute to the assessment of the homogeneity in the high-temperature superconducting tapes. But, before the acceptance, I recommend the authors to address the comment listed below to improve the readability of the paper.

  • It is difficult to read the purpose of this paper. Please mention the purpose more explicitly in the Introduction.

Author Response

Dear reviewer,

We addressed the comment in the text. In the introduction, from line 91, it now reads:

"Through up to date models and their corresponding case studies, the purpose of the present work is to provide an overview of the current developments in the modelling of commercial 2G HTS tapes in the context of power systems, the current needs of the modelling community to achieve the highest accuracy to describe the dynamic behavior of superconducting power devices, and a discussion on the impact of the variability in superconducting properties that may affect the correct operation of those devices as well as the potential issues that are actually impairing its rapid penetration in the national energy grids".

Best regards,

The authors

Reviewer 3 Report

This is a well written paper in which the state of the art of thermoelectric modelling of coated conductors. The paper includes an extensive review of models based on lumped parameters and of FEA approaches. A rigorous treatment of thermal and electrical modelling is performed and the authors successfully revisit the coupling between FEM and the circuit model. This work also highlights some important gaps, such as the empirical data scarcity and the necessary bridge between the superconductivity and power system engineering communities.

However, I find that the review might give a false impression about FEM-circuit coupling model: some assumptions (temperature dependence of some physical phenomena is not included) are typically made, but this is not a limitation of the approach. It does not seem a limitation of the FEM-circuit coupling impossible to overcome. A deeper discussion on the computational cost of each approach would be appreciated. In Section 4 I miss more details on FEM implementation: do the authors use an in-house built code or a commercial software? Since the authors mention computational efforts, did they carry out a study on mesh size influence on time and solution accuracy? Which type of integration has been used?

In any case, the discussion is very convenient and useful for researchers on thermoelectric modelling and the paper is well structured. Additionally, the references included for both modelling approaches are comprehensive.  It is appreciated that the writing style is very fluent and English editing is not required at all.

For all these reasons, I recommend the publication of this paper after minor revision.

The following comments should also be addressed:

  • I find that the first part of the abstract is too long and introductory to the context of REBCO tapes. I would reduce the part up to the sentence “The present work addresses some of the latest developments…” where the aim of the paper is presented.
  • In my opinion, the meaning of the acronym REBCO (Rare-earth barium copper oxide) should be included in the abstract and/or the introduction.
  • When there is a highly non-linear response, as the author states for HTS materials, this mathematical consequence must be associated to a physical process. Could you give more details of these non linearity or any reference? Is it all caused by the nonlinear resistance?
  • What do the author mean by “while balancing the computational time” in line 84? I am not familiar with that modelling. Is the computational time significant to require balancing?
  • From my point of view, the authors present the FEA as a too novel or sophisticated tool. It must be noted that the FEA is an established method for thermal and electromagnetic phenomena. It also must be warned that, as every discretization method, the FEM gives approximated solutions, whose accuracy depends on mesh and optimal choice of solver parameters and time integration. Thus, lines 98 to 100 should be reformulated. The “heaviness” in terms of computational load of coupled thermal and electromagnetic models is rather subjective, but I agree with the statement of the authors about the “usefulness in the design tuning of the superconducting tuning”.
  • Figure 1 should advise of the logarithmic scale (even though it could seem obvious). A scheme of the estimation of temperature effects on the E-J evolution would also be illustrative.
  • I miss a discussion on the thermal capacitances (line 249): temperature dependence, particularities on superconductors…
  • Regarding Section 3.2., references assuming linear dependency of I_C on temperature would be appreciated, as well as a discussion on the possible limitations.
  • For the T-A formulation in FEM I did not understand whether the system (eq. 16 and 17) is solved in a fully coupled or separated (weak coupling) numerical scheme. This difference is interesting when constructing Jacobian matrices and to discuss the computational cost of these models.

Author Response

Dear reviewer,

We addressed all your comments to the best of our knowledge and modified the text were it was due in color red.

We are providing hereinafter our replies. We also submitted a pdf document that may more readable (section Reviewer 3).

best regards,

the authors

Replies:

This is a well written paper in which the state of the art of thermoelectric modelling of coated conductors. The paper includes an extensive review of models based on lumped parameters and of FEA approaches. A rigorous treatment of thermal and electrical modelling is performed and the authors successfully revisit the coupling between FEM and the circuit model. This work also highlights some important gaps, such as the empirical data scarcity and the necessary bridge between the superconductivity and power system engineering communities.

However, I find that the review might give a false impression about FEM-circuit coupling model: some assumptions (temperature dependence of some physical phenomena is not included) are typically made, but this is not a limitation of the approach. It does not seem a limitation of the FEM- circuit coupling impossible to overcome.

  •  A deeper discussion on the computational cost of each approach would be appreciated.

We added information regarding the computation time in the text for the different case studies with a few lines in the discussion section. No comparison is offered since the case studies are widely different but it provides an overview of the computation time in lumped-parameters models and FEM models.

Lines: 309, 426, 545, 581 and in the discussion a few lines

  •  In Section 4, I miss more details on FEM implementation: do the authors use an in-house built code or a commercial software? Since the authors mention computational efforts, did they carry out a study on mesh size influence on time and solution accuracy? Which type of integration has been used?

It uses COMSOL Multiphysics. A current effort is carried out to develop similar codes in open- source software with a particular focus on Onelab (http://onelab.info). Such software demands more developments than in commercial software and has less impact on the community at this stage as the main stream tool in HTS modelling is COMSOL for its ease of use. Indeed, it does not require any knowledge of programming. It is often required to compare the new developments with COMSOL in the HTS modelling community.

We added a few lines in the discussion on the computational time of the different models (lines: 595, 604). The time integration is an adaptive scheme with a BDF order between 1 and 5. The linear system of equations is obtained via a newton-Raphson algorithm and the direct solver MUMPS is used to compute the solution for the system Ax=b.

In any case, the discussion is very convenient and useful for researchers on thermoelectric modelling and the paper is well structured. Additionally, the references included for both modelling approaches are comprehensive. It is appreciated that the writing style is very fluent and English editing is not required at all.

For all these reasons, I recommend the publication of this paper after minor revision.

The following comments should also be addressed:

  •   I find that the first part of the abstract is too long and introductory to the context of REBCO tapes. I would reduce the part up to the sentence “The present work addresses some of the latest developments...” where the aim of the paper is presented.

    The abstract has been slightly modified to reduce its length while keeping enough background to introduce the aim of the paper.

  •   In my opinion, the meaning of the acronym REBCO (Rare-earth barium copper oxide) should be included in the abstract and/or the introduction.

    We added right at the beginning of the abstract where the term REBCO appears the following: “The manufacturing of commercial REBCO tapes, REBCO referring to Rare-earth barium vopper oxide, has matured enough to lead to a variety of applications ranging from scientific instruments to electrical power systems”

  •   When there is a highly non-linear response, as the author states for HTS materials, this mathematical consequence must be associated to a physical process. Could you give more details of these non linearity or any reference? Is it all caused by the nonlinear resistance?

    The physical process is described by Fig. 1 in the text. The superconductor undergoes a change of state going from the superconducting state to the normal-resistive state. The process originates from the movements of the vortices (quantum magnetic fluxes). The motion is triggered either by applying an external increasing magnetic field or a change in the local temperature. Potential wells, at grain boundaries, dislocations, alternate phases, etc..., are typically naturally appearing or artificially created in the superconductor by mechanical and heat treatment to anchor these fluxoids. The network of vortices that forms a lattice when moving generates dissipation in the bulk of the superconductor and the dissipation in turns changes the local temperature and leads to further motion with increasing nucleation of vortices.

    The critical current density is related to the pinning force anchoring those vortices. It demarks the limit between a static lattice and a moving lattice. Creeping motion occurs when the vortices can reach a new configuration after a redistribution. The flow of vortices arising when the equilibrium between the pulling force (“Lorentz force”) is larger than the anchoring one and a rapid increase in dissipation leads to the loss of the superconducting state.

    We added a sentence in the text after the description of Fig. 1 (line 182) and two references on the physics behind the different states spanning the superconducting state to the normal-resistive state for a hard high temperature superconductor: “Further details on the underlying physics of the transition process can be found in [40], [41].“

  •   What do the author mean by “while balancing the computational time” in line 84? I am not familiar

    with that modelling. Is the computational time significant to require balancing?

    Indeed, the idea is to create models that can be run as quickly as possible on a personal computer. It is an underlying goal of the HTS modelling community and a necessity for modelling the device in a more complex network involving the modelling of more items (such as in a power grid). There is a tradeoff between the level of details to be input in the model, its accuracy and the computation time that should be considered if large models at longer simulation times have to be built.

  •   From my point of view, the authors present the FEA as a too novel or sophisticated tool. It must be noted that the FEA is an established method for thermal and electromagnetic phenomena. It also must be warned that, as every discretization method, the FEM gives approximated solutions, whose accuracy depends on mesh and optimal choice of solver parameters and time integration. Thus, lines 98 to 100 should be reformulated. The “heaviness” in terms of computational load of coupled thermal and electromagnetic models is rather subjective, but I agree with the statement of the authors about the “usefulness in the design tuning of the superconducting tuning”.

    We added a couple of sentences to reflect the fact that the FEM model is a numerical method relying on the spatial discretization limiting the accuracy of the results. We have a few references to back up that the FEM has been successfully used in the modelling of HTS tape.

    Line 71: “It should be noted that it is a numerical method relying on the spatiotemporal discretization of partial differential equations for transient analyses. By essence, it leads to an approximated solution which accuracy depends on the level of geometric details and the fineness of the mesh used to discretize spatially the geometry.”

    We agree that it is an established method, which has been strongly developed in the 80’s for electromagnetism. However, we argue that it is not simple to use it in the modelling of HTS tapes and not the least it is not the tool used by power systems people. It requires specific knowledge. It is in this sense sophisticated. Its usage requires new approaches and sometimes new techniques (homogenization, multi-scaling, etc...) to be effectively employed in the modelling of highly non-linear problems such as found in the modelling of superconductors. The problem gets stiff and numerical issues have to be overcome (convergence and memory usage). For instance, it has been reported that the well-known A-V formulation of the Maxwell equation is not the best formulation to tackle the electromagnetic modelling of HTS tapes. The H-formulation with edge elements has been the “de facto” approach to model 2D problems involving HTS tapes over the past 20 years. Very recently (4 years ago), the T-A formulation proved to be the best formulation to move beyond the limitation of the H-formulation. It is currently developed to simulate 3D large scale models. It provides a rapid solution at a high accuracy (very close to the accuracy provided by the H-formulation) (see citation [35] in the manuscript).

    The FEM is an approximation indeed. A direct solver is typically used to improve the convergence as they have a good convergence on highly nonlinear problems. Either a Picard’s scheme or a Newton’s algorithm is applied. The latter being the most common numerical linearization approach. Libraries such as PARDISO and MUMPS are used with high success in COMSOL. The tweaking of the parameters of the solver is always the “magic” part in the modelling of HTS tapes. Some experience is required to get the fastest convergence at the best accuracy.

  •   Figure 1 should advise of the logarithmic scale (even though it could seem obvious). A scheme of the estimation of temperature effects on the E-J evolution would also be illustrative.

    Fig. 1 was modified accordingly. We added the impact of the temperature in the n index and critical current density as well and a line in the text (line 182).

  •   I miss a discussion on the thermal capacitances (line 249): temperature dependence, particularities on superconductors.

    We added more information in the paragraph. Further details can be found in [36].

    From line 235, the paragraph was also modified, and we also added a note in the caption of Fig. 4. We added some more explanations such as: “C is the matrix of thermal capacitances. It is a diagonal matrix made of the heat capacitances of the diverse material associated with their respective temperature nodes accounting for the increase in internal energy during transient dissipations (not shown in Fig. 4). The thermal conductivities, heat capacities and the electrical resistivities ρ build in the resistances of each layers (Rk = ρk*ltp/Ak , A the cross section area of the layer k) depends on temperature according to the data provided by the National Institute of Standards and Technology (NIST) [42]. P is the power dissipated in the tape layers as provided

    by (12), and Dt represent the current time and the time step, respectively. More details of the thermal model including the definition of each term of (13) can be found in [36].”

  • Regarding Section 3.2., references assuming linear dependency of I_C on temperature would be appreciated, as well as a discussion on the possible limitations.

    We added a couple of references. A brief discussion has been added as well.

    Line 209: “It is a common feature of models to assume such linear behavior at 77~K which has been verified experimentally through the measurements of the critical current density as a function of temperature over a wide range of temperatures from 77~K down [35], [41]. However, it is not clear that such a linear feature holds close to the critical temperature [42].”

    The issue is still the lack of experimental data for commercial tapes, which is particularly true for instance between 77 K and the critical temperature.

  •   For the T-A formulation in FEM I did not understand whether the system (eq. 16 and 17) is solved in a fully coupled or separated (weak coupling) numerical scheme. This difference is interesting when constructing Jacobian matrices and to discuss the computational cost of these models.

    It is segregated that provides typically better convergence for multiphysics problems. The equation (18) shows the coupling between both systems and Fig. 19 provides the scheme to couple FEM and circuits.
